# Learning to Reason with Neural Networks: Generalization, Unseen Data and Boolean Measures

**Emmanuel Abbe**[*]
EPFL

**Samy Bengio**
Apple

**Elisabetta Cornacchia**
EPFL

**Jon Kleinberg**
Cornell University

**Aryo Lotfi**
EPFL

**Maithra Raghu**
Google Research

**Chiyuan Zhang**
Google Research

## Abstract

This paper considers the Pointer Value Retrieval (PVR) benchmark introduced in [ZRKB21], where a 'reasoning' function acts on a string of digits to produce the label. More generally, the paper considers the learning of logical functions with gradient descent (GD) on neural networks. It is first shown that in order to learn logical functions with gradient descent on symmetric neural networks, the generalization error can be lower-bounded in terms of the *noise-stability* of the target function, supporting a conjecture made in [ZRKB21]. It is then shown that in the distribution shift setting, when the data withholding corresponds to freezing a single feature (referred to as canonical holdout), the generalization error of gradient descent admits a tight characterization in terms of the *Boolean influence* for several relevant architectures. This is shown on linear models and supported experimentally on other models such as MLPs and Transformers. In particular, this puts forward the hypothesis that for such architectures and for learning logical functions such as PVR functions, GD tends to have an *implicit bias towards low-degree representations*, which in turn gives the Boolean influence for the generalization error under quadratic loss.

## 1 Introduction

Recently [ZRKB21] introduced the pointer value retrieval (PVR) benchmark. This benchmark consists of a supervised learning task on MNIST [LCB10] digits with a 'logical' or 'reasoning' component in the label generation. More specifically, the functions to be learned are defined on MNIST digits organized either sequentially or on a grid, and the label is generated by applying some 'reasoning' on these digits, with a specific digit acting as a pointer on a subset of other digits from which a logical/Boolean function is computed to generate the label.

For instance, consider the PVR setting for binary digits in the string format, where a string of MNIST digits is used as input. Consider in particular the case where only 0 and 1 digits are used, such as in the example of Figure 1. The label of this string is defined as follows: the first 3 bits 101 define the pointer in binary expansion, and the pointer points to a window of a given length, say 2 in this example. Specifically, the pointer points at the first bit of the window. To generate the label, one has

---

[*]Authors are in alphabetical order.

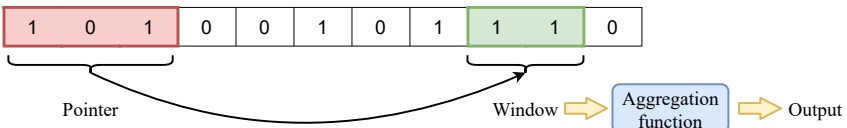

Figure 1: An example of a PVR function with a window size of 2. The first 3 bits are the pointer, which points to a window in the subsequent bits. Specifically, the number indicated by the pointer bits in binary expansion gives the position of the first bit of the window. The label is then produced by applying some fixed aggregation function to the window bits (e.g., parity, majority-vote, etc.).

to thus look the 6th window[2] of length 2 given by 11, and there the label is produced by applying some fixed function, such as the parity (so the label would be 0 in this example). In [ZRKB21], the PVR benchmark is also defined for matrices of digits rather than strings; we focus here on the string version that captures all the purposes of the PVR study. This benchmark is introduced to understand the limits of deep learning on tasks that go beyond classical image recognition, investigating in particular the trade-off between memorization and reasoning by acting with a particular distribution shift at testing (see further details below).

In order to learn such PVR functions, one has to first learn the digit identification and then the logical component on these digits. Handling both tasks successfully at once is naturally more demanding than succeeding at the latter assuming the first is successful. Here, we focus on the 'logical component' as a necessary component to learn, and this corresponds to learning a Boolean function. The overall function that maps the pixels of an image to its label in the PVR is of course also a Boolean function (like any computer-encoded function), but the structural properties of such meta-functions are more challenging to describe, and left for future work. In any case, to understand the limits of deep learning on such benchmarks, we focus on investigating first the limits of deep learning on the logical/Boolean component.

We next re-state formally one of the PVR benchmarks from [ZRKB21], focusing on binary digits for simplicity. We will use the alphabet $\{0, 1\}$ to describe the benchmark to connect to the MNIST dataset, but will later switch to the alphabet $\{+1, -1\}$ for the problem of learning Boolean functions with neural networks. Recall for $d \in \mathbb{N}$, $\mathbb{F}_2^d = \{0, 1\}^d$.

**Definition 1** (Boolean PVR with *sliding* windows). *The input $x$ consists of $n$ bits and the label is generated as*

$$f(x_1, \ldots, x_n) = g(x_{P(x^p)}, \ldots, x_{P(x^p)+w-1}). \tag{1}$$

*where $p$ is the number of bits in the pointer, $w$ is the window size, $g : \mathbb{F}_2^w \to \mathbb{R}$ is the aggregation function, $P : \mathbb{F}_2^p \to [p+1 : n]$ is the pointer map, $x_j$ denotes the bit in position $j$ and $x^p = (x_1, ..., x_p)$ denotes the pointer bits. We often set $n = p + 2^p$, and hence the last window starts at the last digit.*

*In words, the first $p$ bits give a pointer to the beginning of a window of $w$ consecutive bits, and the label is produced by applying a Boolean function $g$ on these $w$ bits.*

*Remark* 1. If $w > 1$, for some values of the pointer, $P(x^p) + w - 1$ would exceed the dimension of input $n$. This issue can be solved by using cyclic indices or by using non-overlapping windows as defined in Appendix D. However, in the experiments of this paper, we mainly truncate windows (if necessary) in order to capture the underlying asymmetries (e.g., the last windows have smaller sizes).

In this paper we consider the problem of learning in the holdout setting, i.e., when some data are withheld (or 'unseen') during training. We focus on a specific type of holdout setting, that we call 'canonical holdout', which we define here. Extensions to other types of holdout are also of interest, and left for future work.

**Definition 2** (Canonical holdout). *Let $\mathcal{F}$ be the class of Boolean functions on $n$ bits and let $f$ be a specific function in that class. For $k \in [n]$, consider the problem of learning $\mathcal{F}$ from samples $(X_{-k}, f(X_{-k}))$, where the $X_{-k}$'s are independently drawn from the distribution that freezes component $k$ to 1 and that draws the other components i.i.d. Bernoulli$(1/2)$. Let $\tilde{f}_{-k}$ be the function*

---

[2]pointer 000 points at the first window, pointer 001 at the second window, and so on. Thus pointer 101, that is equal to 5 in binary expansion, points at the 6th window.

*learned under this training distribution. We are interested in the generalization error with square loss when the $k$-th bit is not frozen at testing, i.e.,*

$$\text{gen}(\mathcal{F}, \tilde{f}_{-k}) = \frac{1}{2}\mathbb{E}_{X \sim_U \mathbb{F}_2^n, f \sim_U \mathcal{F}}(f(X) - \tilde{f}_{-k}(X))^2, \tag{2}$$

*where by $X \sim_U \mathbb{F}_2^n$ we mean that $X$ is chosen uniformly at random from $\mathbb{F}_2^n$, and similarly for $f \sim_U \mathcal{F}$. We denote by $\tilde{f}$ the function learned in the case where the $k$-th component is not frozen at training, i.e., when the train and test distributions are both uniform on the Boolean hypercube, and use the following notation for the corresponding generalization error:*

$$\text{gen}(\mathcal{F}, \tilde{f}) = \frac{1}{2}\mathbb{E}_{X \sim_U \mathbb{F}_2^n, f \sim_U \mathcal{F}}(f(X) - \tilde{f}(X))^2. \tag{3}$$

The 'canonical' holdout is thus a special case of holdout where a single feature/bit is frozen at training. In [ZRKB21], different types of holdout are allowed where a given string is absent from the windows rather than a given bit as in the canonical version. We believe that the canonical holdout still captures the essence of the PVR benchmark: other windows will get to see the strings that are withheld due to the bit freezing, and thus the goal is still to investigate whether neural networks trained by GD will manage to patch together the different pieces.

Finally, we will often consider neural network architectures that have invariances on the input, such as permutation invariance. In the PVR setting, this means that we do not assume that the learner has knowledge of which bits are in the window. In such cases, instead of learning a class of function $\mathcal{F}$, one can equivalently talk about learning a single function $f$ (with the implicit $\mathcal{F}$ defined as the orbit of $f$ through the permutation group), and define

$$\text{gen}(f, \tilde{f}_{-k}) = \frac{1}{2}\mathbb{E}_{X \sim_U \mathbb{F}_2^n}(f(X) - \tilde{f}_{-k}(X))^2 \quad \text{and} \quad \text{gen}(f, \tilde{f}) = \frac{1}{2}\mathbb{E}_{X \sim_U \mathbb{F}_2^n}(f(X) - \tilde{f}(X))^2.$$

## 1.1 Contributions of this paper

The contributions of this paper are:

1. In the matched setting (i.e., train and test distributions are matching), we prove a lower-bound (Theorem 1) on the generalization error for gradient descent[3] (GD), that degrades for functions having large *noise-sensitivity* (or low noise-stability), supporting thereby with a formal result a conjecture put forward in [ZRKB21].

2. In the mismatched setting, specifically in the canonical holdout where a single feature is frozen at training and released to be uniformly distributed at testing, we hypothesize that *(S)GD on the square loss[4] and on certain network architectures such as MLPs and Transformers has an implicit bias towards low-degree representations when learning logical functions such as Boolean PVR functions.* This gives a new insight into the implicit bias study of GD on neural networks for the case of logical functions.

   We then show (Lemma 1) that under this hypothesis, the generalization error in the canonical holdout setting is given by the Boolean influence, a central notion in Boolean Fourier analysis.

3. We provide experiments supporting this hypothesis for various target functions and architectures such as MLPs and the Transformers [VSP+17] in Section 3.2 and Appendix F. These rely on mini-batch stochastic gradient descent (see Section 3.2 for more details).

4. We establish formally the hypothesis for GD and linear regression models in Section 3.3, and conduct experiments to support it on multi-layer neural networks of large depths and small initialization scales in Section 3.3 (with further arguments in Appendix G).

## 1.2 The low-degree bias hypothesis: illustration

We now discuss the hypothesis put forward in this paper that "GD on certain architectures such as MLPs and Transformers has an implicit bias towards lower-degree representations when learning

---

[3]This part relies on population gradients with polynomial gradient precision as in [AS20, AKM+21]

[4]We do not expect the square loss to be critical, but it makes the connection to the influence more explicit.

PVR and, more generally, logical functions." Before starting our illustration let us recall some basic notions from Boolean analysis (we refer to [O'D14] for details). Any Boolean function $f : \{\pm 1\}^n \to \mathbb{R}$ can be written in terms of its Fourier-Walsh transform: $f(x) = \sum_{T \subseteq [n]} \hat{f}(T) \chi_T(x)$, where $\chi_T(x) = \prod_{i \in T} x_i$ and $\hat{f}(T) = 2^{-n} \sum_{x \in \{\pm 1\}^n} f(x) \chi_T(x)$ are respectively the basis elements and the coefficients of the Fourier-Walsh transform of $f$.

**Definition 3** (Boolean influence [O'D14]). *Let $f : \{\pm 1\}^n \to \mathbb{R}$ be a Boolean function and let $\hat{f}$ be its Fourier-Walsh transform. The Boolean influence of variable $k \in [n]$ on $f$ is defined by $\mathrm{Inf}_k(f) := \sum_{T \subseteq [n]: k \in T} \hat{f}(T)^2$. In particular, if $f : \{\pm 1\}^n \to \{\pm 1\}$, $\mathrm{Inf}_k(f) = \mathbb{P}(f(X) \neq f(X + e_k))$, where $X + e_k$ corresponds to the vector obtained by flipping the $k$-th component of $X$.*

Consider the following example of a PVR function with a 1-bit pointer, 2 overlapping windows of length 2, and parity for the aggregation function. We consider $f : \{\pm 1\}^n \to \{\pm 1\}$ (i.e., with $\pm 1$ variables instead of $0, 1$, to simplify the expressions), with $f$ given by

$$f(x_1, x_2, x_3, x_4) = \frac{1 + x_1}{2} x_2 x_3 + \frac{1 - x_1}{2} x_3 x_4. \tag{4}$$

We can rewrite $f$ in terms of its Fourier-Walsh expansion (i.e., pulling out all the multivariate monomials appearing in the function, see Section 3.1 for more details), which gives

$$f(x_1, x_2, x_3, x_4) = \frac{1}{2} x_2 x_3 + \frac{1}{2} x_3 x_4 + \frac{1}{2} x_1 x_2 x_3 - \frac{1}{2} x_1 x_3 x_4. \tag{5}$$

Consider now training a neural network such as a Transformer as in Section 3.2 on this function, with quadratic loss, and a canonical holdout corresponding to freezing $x_2 = 1$ at training. Under this holdout, and under the 'low-degree implicit bias' hypothesis, the low-degree monomials are learned first (see experiments in Section 3.2), resulting in the following function being learned at training:

$$f_{-2}(x_1, x_2, x_3, x_4) = \frac{1}{2} x_3 + \frac{1}{2} x_3 x_4 + \frac{1}{2} x_1 x_3 - \frac{1}{2} x_1 x_3 x_4 = \frac{1 + x_1}{2} x_3 + \frac{1 - x_1}{2} x_3 x_4. \tag{6}$$

Thus, according to Lemma 1 proved in Appendix C, the generalization error is given by

$$\frac{1}{2} \mathbb{E}(f(X) - f_{-2}(X))^2 = \mathbb{P}(f(X) \neq f(X + e_2)) = \frac{1}{2}, \tag{7}$$

which is the probability that flipping the frozen coordinate changes the value of the target function, i.e., the Boolean influence (where we denoted by $X + e_2$ the vector obtained by flipping the second entry in $X$). As shown in this paper, neural networks tend to follow this trend quite closely, and we can prove this hypothesis on simple linear models. Notice that an ERM-minimizing function could have taken a more general form than a degree minimizing function, i.e.,

$$f_{-2}^{\mathrm{ERM}}(x) := \frac{1 + x_2}{2} f_{-2}(x) + \frac{1 - x_2}{2} r(x) \tag{8}$$

for any choice of $r : \{\pm 1\}^4 \to \{\pm 1\}$. In the special case of $r = 0$, $f_{-2}^{\mathrm{ERM}}$ corresponds to $f_{-2}$ (the low-degree representation). For instance, among such ERM-minimizers, one can check that the minimum $\ell_2$-norm interpolating solution would be given by

$$f_{-2}^{\ell_2}(x) := \frac{1}{4}(x_3 + x_2 x_3) + \frac{1}{4}(x_3 x_4 + x_2 x_3 x_4) + \frac{1}{4}(x_1 x_3 + x_1 x_2 x_3) - \frac{1}{4}(x_1 x_3 x_4 + x_1 x_2 x_3 x_4). \tag{9}$$

This gives a generalization error of $\frac{1}{2} \mathbb{E}(f(X) - f_{-2}^{\ell_2}(X))^2 = 4(\frac{1}{4})^2 = \frac{1}{4}$, i.e., half the error of $f_{-2}$, yet still bounded away from 0.

In order to improve on this, under the same canonical holdout with $x_2 = 1$, one would like to rely on a type of minimum description length bias, since describing $f$ may be more efficient than $f_{-2}$ due to the stronger symmetries of $f$. Namely, $f$ corresponds to taking the parity on the middle two bits if $x_1 = 1$, and on the last two bits otherwise. On the other hand, $f_{-2}$ requires changing the function depending on $x_1 = 1$ or $x_1 = -1$, since it is once the function $x_3$ and once the function $x_3 x_4$. So an implicit bias that would exploit such symmetries, featuring in PVR tasks, would give a different solution than the low-degree implicit bias, and could result in lower generalization error. We leave to future work to investigate this 'symmetry compensation' procedure.

## 1.3 Related literature

**GD lower bounds.** Several works have investigated the difficulties of learning a class of functions with descent algorithms and identified measures for the complexity in this setting. In particular, [BFJ+94, Kea98] prove that the larger the statistical dimension of a function class is, the more difficult it is for an statistical query (SQ) algorithm to learn. We refer to [AKM+21, DGKP19] for further references on SQ-like lower bounds. For GD with mini-batch and polynomial gradient precision, [AS20] uses the $m$-Cross-Predictability (CP) of a class of functions to show that classes sufficiently small CP and large batch-size are not efficiently learnable. This is further generalized in [AKM+21] to a broader range of batch size and gradient precision. In [AGNZ18], the noise-stability for deep neural networks is defined as the stability of each layer's computation to noise injected at lower layers, and is used to show correctness of a compression framework. However, no bounds on the generalization error of GD depending on the noise stability of the target function is derived in this work. The noise-stability, the statistical dimension and the cross-predictability are measures for some given function or function class. One can obtain measures that are defined for a dataset and a network architecture. For that purpose, [ACHM22] introduced the "Initial Alignment" (INAL) between the target function and the neural network at initialization and proves that for fully connected networks with Gaussian i.i.d. initialization, if the INAL is negligible, then GD will not learn efficiently. We remark that the problem of approximating and learning Boolean functions appear in other areas as well. For instance, [HPV19] considered the problem of approximating Boolean functions, using functions coming from restricted classes (namely k-juntas and linear Boolean functions); [JSR+19] proposes two algorithms to learn sparse Boolean formulae; and [Udo21] proposes techniques for modeling Boolean functions by mixed-integer linear inequalities.

**Implicit bias.** The implicit bias of neural networks trained with gradient descent has been extensively studied in recent years [NTS14, NBMS17, MGW+20, JT19, GLSS18a]. In particular, [SHN+17] proved that gradient descent on linearly-separable binary classification problems converges to the maximum $\ell_2$-margin direction. Several subsequent works studied the implicit bias in classification tasks on various networks architectures, e.g., homogeneous networks [LL19], two layers networks in the mean field regime [CB20], linear and $\mathrm{ReLU}$ fully connected networks [VSS21], and convolutional linear networks [GLSS18b]. Among regression tasks, the problem of implicit bias has been analysed for matrix factorization tasks [GWB+17, RC20, ACHL19], and also gradient flow on diagonal networks [PPVF21]. However, all these works consider functions with real inputs, instead of logical functions which are the focus of this work. On the other hand, as discussed in Section 1.2, the Boolean influence generalization characterization reflects the implicit bias of GD on neural networks to learn low-degree representations. Similar types of phenomena can implicitly be found in [XZX18, XZL+19, RBA+19], in particular as the "spectral bias" in the context of real valued functions decomposed in the classical Fourier basis (where the notion of lower degree is replaced by low frequencies). In [ABAB+21, ABAM22], the case of Boolean functions is considered as in this paper, and it is established that for various 'regular' architectures (having some symmetry in their layers), gradient descent can learn target functions that satisfy a certain 'staircase' property. However, these papers do not investigate lower-bounds in terms of noise stability, neither distribution shift as in the canonical holdout. We note that, in the context of Boolean functions, the problem of regularizing a learning algorithm to avoid overfitting appears in other areas of research, e.g., in the analysis of fitness functions in biology [ANO+21].

**Distribution shift.** Many works were aimed at characterizing when a classifier trained on a training distribution (also called the "source" distribution) performs well on a different test domain (also called the "target" distribution) [QCSSL09]. On the theoretical side, [BDBC+10] obtains a bound of the target error of a general empirical risk minimization algorithm in terms of the source error and the divergence between the source and target distribution, in a setting where the algorithm has access to a large dataset from the source distribution and few samples from the target distribution. We refer to [SQZY17] for a further result in a similar setting. Instead, in this work we focus on gradient descent on neural networks in the setting where no data from the target distribution is accessible. On the empirical level, several benchmarks have been proposed to evaluate performance for a wide range of models and distribution shifts [WGS+22, MTR+21, SKL+21]. Despite this significant body of works on distribution shift, we did not find works that related the generalization error under holdout shift in terms of the Boolean influence.

## 2   Matched setting: a formal lower-bound on noise stability and generalization

Our first result provides a lower bound on the generalization error in the setting where the training and test distributions of the inputs are both uniform on the Boolean hypercube, i.e., the "matched setting". We directly relate the generalization error achieved by GD, or SGD with large batch-size, to the complexity of the task, providing theoretical support to the empirical claim in [ZRKB21], that complex tasks are more difficult to learn for GD/SGD on neural networks. The latter work used the noise sensitivity as a dual measure of the target function complexity, whereas we use here the noise stability ($\mathrm{Stab}_\delta[f]$).

**Definition 4** (Noise stability). *Let $f : \{\pm 1\}^n \to \mathbb{R}$ and $\delta \in [0, 1/2]$. Let $X$ be uniformly distributed on the Boolean $n$-dimensional hypercube, and let $Y$ be formed from $X$ by flipping each bit independently with probability $\delta$. We define the $\delta$-noise stability of $f$ by $\mathrm{Stab}_\delta[f] := \mathbb{E}_{(X,Y)}[f(X) \cdot f(Y)]$.*

Intuitively, $\mathrm{Stab}_\delta[f]$ measures how stable the output of $f$ is to a perturbation that flips each input bit with probability $\delta$, independently. The noise stability can be easily related to the noise sensitivity $\mathrm{NS}_\delta[f]$[5], used in [ZRKB21].

The generalization error depends as well on the complexity of the network, which is quantified in terms of the number of edges in the network, the number of time steps, and gradients precision in the gradient descent algorithm. We give one more definition before stating our result.

**Definition 5** (N-Extension). *For a function $f : \mathbb{R}^n \to \mathbb{R}$ and for $N > n$, we define its $N$-extension $\bar{f} : \mathbb{R}^N \to \mathbb{R}$ as $\bar{f}(x_1, x_2, ..., x_n, x_{n+1}, ..., x_N) = f(x_1, x_2, ..., x_n)$.*

**Theorem 1.** *Consider a fully connected neural network of size $E$ with initialization that is target agnostic in the sense that it is invariant under permutations of the input[6]. Let $f : \{\pm 1\}^n \to \{\pm 1\}$ be a balanced target function. Let $\bar{f}$ be the $2n$-extension of $f$ as defined in Definition 5. Let $f_{\mathrm{NN}}^{(t)}$ be the output of noisy-GD with gradient range $A$, batch-size $b$, learning rate $\gamma$ and noise scale $\sigma$ (see Def. 8) after $t$ time steps. Then, for $\delta$ small enough[7] and for $b$ large enough[8], the generalization error satisfies*

$$\mathrm{gen}\left(\bar{f}, f_{\mathrm{NN}}^{(t)}\right) \geq 1/2 - \frac{\gamma t \sqrt{E} A}{\sigma} \cdot \mathrm{Stab}_\delta[f]^{1/4}. \tag{10}$$

Theorem 1 states a lower bound for learning in the extended input space. We use the input doubling to guarantee that the hypothesis class $\mathcal{F}$ resulting from the orbit of $\bar{f}$, i.e., $\mathrm{orb}(\bar{f}) = \{\bar{f} \circ \pi : \pi \in S_{2n}\}$, is not degenerate to a single function, as this could then be learned using a proper choice of the initialization (i.e., one can simply set the weights of the neural network at initialization to represent the unique function, if the network has enough expressivity). Instead, the input doubling prohibits such representation shortcut and ensures that the structural properties of the function is what creates the difficulty of learning, irrespective of the choice of the initialization. For instance, consider the full parity function on all of the input bits. The orbit of this function contains only that specific function, and one can learn that function by choosing a proper initialization on a neural net of depth 2. However, with an input doubling, this function becomes hard to learn *no matter what* the initialization is [Kea98, AS20]. One can remove this input doubling requirement by assuming that $f$ is non-extremal (i.e., no terms of degree $\theta(n)$ in the Fourier basis) and non-dense (i.e., $\mathrm{poly}(n)$-sized Fourier spectrum), see Appendix B.

The proof of Theorem 1 uses [AS20] which obtains a lower-bound in terms of the cross-predictability (CP) (instead of the noise stability). The CP is a complexity measure of a *class* of functions, rather than a single function. For the orbit of $\bar{f}$, the CP is defined as $\mathrm{CP}(\mathrm{orb}(\bar{f})) = \mathbb{E}_{\pi \sim_U S_{2n}}[\mathbb{E}_{X \sim_U \mathbb{F}_2^{2n}}[\bar{f}(X) \cdot \bar{f} \circ \pi(X)]^2]$. We give a more general definition of CP in Appendix A (we believe that the CP also extends to other invariances than permutations and non i.i.d. distributions).

---

[5]Specifically, for binary-valued functions, $\mathrm{NS}_\delta[f] = \frac{1}{2} - \frac{1}{2}\mathrm{Stab}_\delta[f]$.

[6]One could consider other groups of invariances than the full permutation group; this is left for future work.

[7]Generally, this holds for any $\delta$ such that $CP(\mathrm{orb}(\bar{f})) \leq \mathrm{Stab}_\delta[f]$; in particular, this holds for $\delta < 1/4$ under input doubling or for some $\delta > 0$ under non-extremal and non-dense assumptions.

[8]This holds for $b \geq 1/CP(\mathrm{orb}(\bar{f}))$.

Theorem 1 states that if the target function is highly noise unstable, specifically if there exists $\delta < 1/4$ such that $\mathrm{Stab}_\delta[f]$ decreases faster than any inverse polynomial in $n$, then GD will not learn the $2n$-extension of $f$ (or $f$ itself if it is non-extremal/dense) in polynomial time and with a polynomially sized neural network initialized at random. So in that sense, the noise-stability gives a proxy to generalization error, as observed in [ZRKB21]. More specifically, this result is about failure of the weakest form of learning no matter what the architecture is. One could also consider 'regular' architectures (such as with isotropic layers) and stronger learning requirements; for this it is known that the 'Fourier leap' from [ABAB$^+$21] is a relevant complexity measure, and we leave investigations of regular architectures to future work. More details of the proof of Theorem 1 are in Appendix A. In Appendix D, we explain how the noise stability of PVR functions can be computed, and the implications to the result of Theorem 1.

## 3 Mismatched setting

In this section, we focus on the generalization error under the distribution shift setting, more specifically the canonical holdout setting defined in Definition 2. Namely, assume that at training component $k$ is frozen to 1, and assume it to be released to $\mathrm{Unif}\{\pm 1\}$ at testing. Our experiments show that in this setting, for some relevant architectures, the generalization error is close to the value of a standard measure in Boolean analysis, namely the Boolean influence of variable $k$ on $f$.

### 3.1 Boolean influence and generalization

To explain the connection between generalization error and Boolean influence, we start with a simple lemma relating the Boolean influence to the $\ell_2$-distance between the true target function and the function obtained by freezing component $k$ (which we call the "frozen function").

**Lemma 1.** *Let $f : \{\pm 1\}^n \to \mathbb{R}$ be a Boolean function and let $f_{-k}$ be defined as $f_{-k}(x) := f(x_{-k})$ where $x_{-k}(i) = 1$ if $i = k$ and $x_{-k}(i) = x(i)$ otherwise. Then, $\frac{1}{2}\mathbb{E}_X(f(X) - f_{-k}(X))^2 = \mathrm{Inf}_k(f)$.*

The proof of Lemma 1 can be found in Appendix C. In Section 3.2, we present experiments that demonstrate the relation between the Boolean influence and the generalization error for different architectures. In Section 3.3, we focus on linear models, namely, linear regression and linear networks.

### 3.2 Experiments

We consider three architectures for our experiments: multi-layer perceptron (MLP) with 4 hidden layers, the Transformer [VSP$^+$17], and MLP-Mixer [THK$^+$21]. For each architecture, we freeze different coordinates separately and evaluate our models. In other words, we train the model while freezing coordinate 1, then coordinate 2 and so on, until coordinate $n$ and compare the generalization error with the Boolean influence of the corresponding coordinates of the target function. We train our models using $\ell_2$ loss and mini-batch SGD with momentum and batch-size of 64 as the optimizer. Moreover, we have repeated each experiment 40 times and averaged the results. Furthermore, note that for the MLP model, we pass the Boolean vector directly to the model. However, for the Transformer and MLP-Mixer, we first encode $+1$ and $-1$ tokens into 256-dimensional vectors using a shared embedding layer, and then we pass the embedded input to the models. More details on training procedure as well as further experiments are presented in Appendix F.[9]

**Influence vs. canonical holdout generalization.** In this section, we consider a Boolean PVR function (as in Def. 1) with 3 pointer bits to be learned by the neural networks. For this function we set window size to 3 and use majority-vote (defined as $g(x_1, \ldots, x_r) = \mathrm{sign}(x_1 + \cdots + x_r)$, outputting $-1$, $0$, or $1$) as the aggregation function on the windows. The generalization error of the models on this PVR function and its comparison with the Boolean influence are presented in Figure 2. The x-axis corresponds to the index of the frozen coordinate, that is from 4 to 11 (we do not freeze the pointer bits). On the y-axis, for each frozen coordinate we report the generalization error obtained in the corresponding holdout setting for MLP, the Transformer, and MLP-Mixer, together with the value of the Boolean influence of the frozen coordinate on the target function. It can be seen that the generalization error of MLP and the Transformer can be well approximated by the Boolean influence.

---

[9]Code: https://github.com/aryol/BooleanPVR

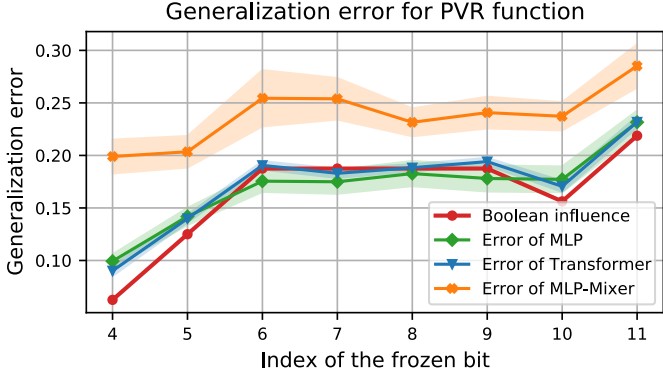

Figure 2: Comparison between the generalization loss in the canonical distribution shift setting and the Boolean influence for a PVR function with 3 pointer bits, window size 3, and majority-vote aggregation function. See Appendix F for further experiments.

Whereas, the generalization error of MLP-Mixer follows the trend of the Boolean influence with an offset. We remark that in Figure 2, the value of the Boolean influence (and gen. error) in the PVR task varies across different indices due to boundary effects and the use of truncated windows (see Def. 1). We refer to Appendix F for further experiments on other target functions.

**Implicit bias towards low-degree representation.** Consider the problem of learning a function $f$ in the canonical holdout setting freezing $x_k = 1$. Denote the Fourier coefficients of the frozen function $f_{-k}$ (as defined in Lemma 1), by $\hat{f}_{-k}(S)$, for all $S \subseteq [n] := \{1, ..., n\}$ (recall, $\hat{f}_{-k}(S) = \mathbb{E}_X[f_{-k}(X)\chi_S(X)]$). For $S$ such that $k \notin S$, the neural network can learn coefficient $\hat{f}_{-k}(S)$ using either $\chi_S(x)$ or $x_k \cdot \chi_S(x) = \chi_{S \cup \{k\}}(x)$ (since these are indistinguishable at training). The low-degree implicit bias states that neural networks have a preference for the lower degree monomial $\chi_S$. More precisely, $\chi_S$ is learned faster than $\chi_{S \cup \{k\}}$ and thus the term $\hat{f}_{-k}(S)\chi_S(x)$ in the Fourier expansion of $f_{-k}$, is mostly learned by the lower degree monomial. Consequently, according to Lemma 1, the generalization error will be close to the Boolean influence. Figure 3 shows this bias empirically for the above mentioned PVR function and for frozen coordinate $k = 6$. Figure 3 (left) shows that the MLP model has a strong preference for low-degree monomials, Similarly, Figure 3 (bottom) shows that the Transformer also captures the monomials in the original function using monomials that exclude the frozen index. Therefore the generalization errors of the MLP and Transformer are very close to the Boolean influence as seen in Figure 2. Whereas, Figure 3 (right) shows that the MLP-Mixer model has a weaker preference for lower degree monomials (e.g., it learns 1 and $x_6$ to the same extent) and hence, its generalization error follows the trend of Boolean influence with an offset, which is also presented in Figure 2. We refer to Appendix F for additional experiments.

### 3.3 Linear Models

In this section, we will focus on linear functions and linear models. First, we state a theorem for linear regression models. Furthermore, we show experiments on the generalization error of linear neural networks and its relation with the depth and initialization of the model.

**Theorem 2.** *Let $f : \{\pm 1\}^n \to \mathbb{R}$ be a linear function, i.e., $f(x_1, \cdots, x_n) = \hat{f}(\emptyset) + \sum_{i=1}^n \hat{f}(\{i\})x_i$. Consider the canonical holdout where the $k$-th component is frozen at training for a linear regression model where weights and biases are initialized independently with the same mean and variance $\sigma^2$. Also assume the frozen function is unbiased, i.e., $\mathbb{E}_{X_{-k}}[f(X)] = 0$. In this case, the expected generalization error (over different initializations) of the function learned by GD after $t$ time steps is given by $\mathbb{E}_{\Theta^0}[\text{gen}(f, \tilde{f}_{-k}^{(t)})] = \text{Inf}_k(f) + o_{\sigma^2}(1) + O(e^{-ct})$, where $c$ is a constant dependent on the learning rate. Moreover, if the frozen function is biased, the expected generalization error is equal to $\mathbb{E}_{\Theta^0}[\text{gen}(f, \tilde{f}_{-k}^{(t)})] = \frac{(\hat{f}(\emptyset) - \hat{f}(\{k\}))^2}{4} + o_{\sigma^2}(1) + O(e^{-ct})$.*

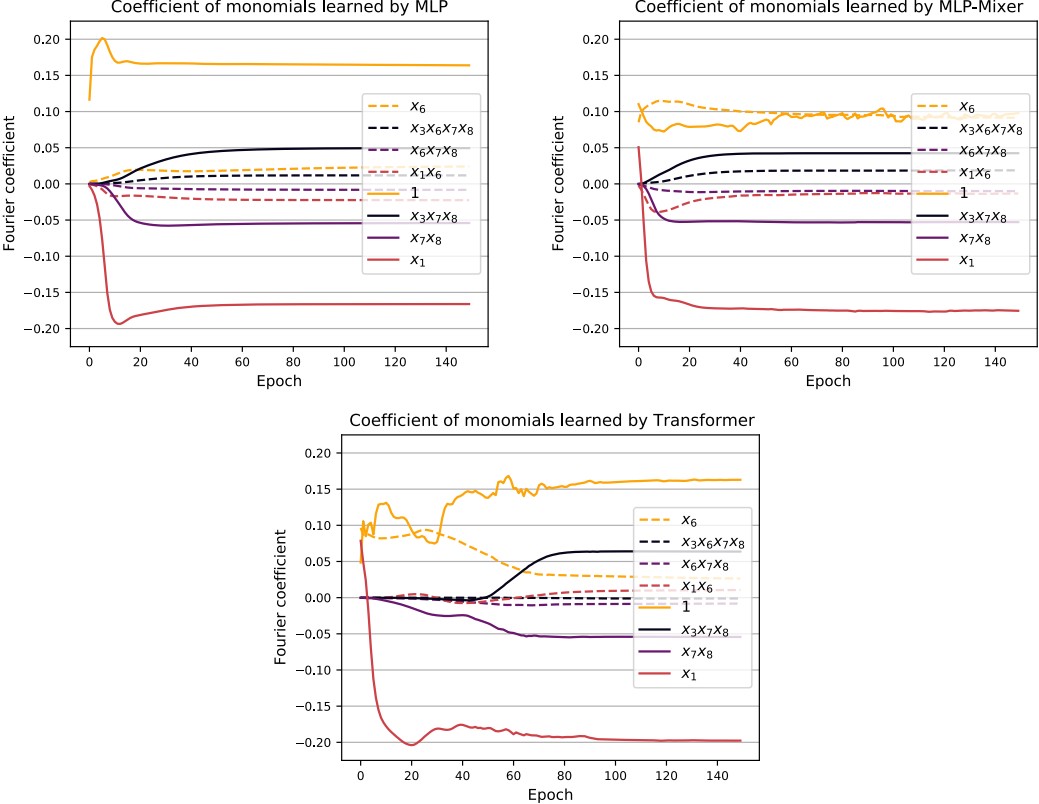

Figure 3: The coefficients of a selected group of monomials learned by the MLP (left), MLP-Mixer (right) and the Transformer (bottom) when learning the aforementioned PVR function, with $x_6 = 1$ frozen during the training. The coefficient of these monomials in the original function are $\hat{f}(\{6\}) = 0.1875$, $\hat{f}(\{3,6,7,8\}) = 0.0625$, $\hat{f}(\{6,7,8\}) = -0.0625$, and $\hat{f}(\{1,6\}) = -0.1875$. One can observe that the monomials of the lowest degree are indeed picked up first during the training of the MLP and Transformer, which explains the tight approximation of the Boolean influence for the generalization error in these cases. In contrast, the MLP-Mixer also picks up some contribution from the higher degree monomials including the frozen bit $x_6$.

*Remark* 2 (Kernel regression). Our result would still hold if, instead of linear regression, we performed kernel regression with a kernel $K$ that is invertible under the training distribution, i.e., such that $\mathbb{E}_{X,X' \sim \mathcal{U}_{-k}}[K(X, X')]$ is invertible. Furthermore, one could try to remove the invertibility assumption by adding a regularization term.

*Remark* 3 (Staircase learning). An attempt to extend the above result to non-linear function consists of considering staircase functions [ABAB+21, ABAM22], and extending [ABAM22] to a setting with bias in order to show that SGD learns the lowest-degree representation under proper mean-field initialization of depth-2 neural networks.

The proof of Theorem 2 is presented in Appendix C. In the rest of this section, we empirically show that the condition of zero bias stated in Theorem 2 is no longer necessary if linear neural networks of large enough depth or small enough initialization are used. In fact, the generalization error of linear neural networks makes a transition from the value proved in Theorem 2 to the Boolean influence as deeper models or smaller scales of initialization are used. We take $f(x_1, x_2, \ldots, x_{11}) = 1 + 2x_1 - 3x_2 + 4x_3 - \cdots - 11x_{10} + 12x_{11}$ as the function that we want to learn. We consider linear neural networks with hidden layers of size 256. Figure 4 (left) shows the effect of depth: we initialize weights and biases independently using the uniform distribution $\mathcal{U}(-\frac{1}{\sqrt{N_{in}}}, \frac{1}{\sqrt{N_{in}}})$ where $N_{in}$ is the input dimension of the respective layer. We plot the generalization error in the holdout setting with respect to the corresponding frozen coordinate at training, together with the value of the Boolean influence of each coordinate. We observe that with the increase of depth, the generalization error tends to the Boolean influence. Figure 4 (right) show the role of initialization: we take a linear neural network with 3 layers and we initialize weights and biases using the uniform distribution

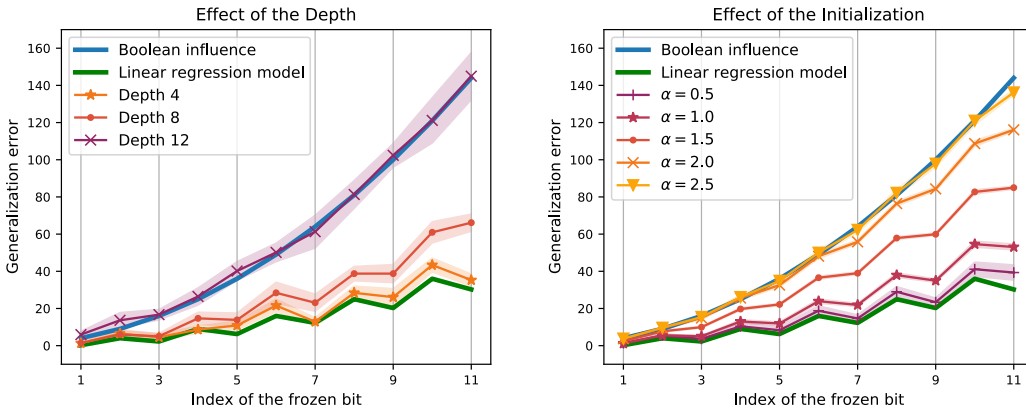

Figure 4: Effect of depth (left) and initialization scale (right) on the out-of-distribution generalization error of linear neural networks. Generalization error tends to the Boolean influence as depth increases (left) or as the initialization scale decreases (right).

$\mathcal{U}(-N_{in}^{-\alpha}, N_{in}^{-\alpha})$ with $\alpha$ taking value in $\{0.5, 1, 1.5, 2, 2.5\}$. It can be seen that as the initialization scale decreases, the generalization error tends to the Boolean influence.

## 4    Conclusion

This paper first establishes a formal result that supports a conjecture made in [ZRKB21], relating the noise sensitivity of a target function to the generalization error. This gives a first connection between a central measure in Boolean analysis, the noise-sensitivity, and the generalization error when learning Boolean functions with GD. The paper then investigates the generalization error under the canonical holdout. The 'low-degree implicit bias hypothesis' is put forward and supported both experimentally and theoretically for certain architectures. This gives a new insight on the implicit bias of GD when training neural networks, that is specific to Boolean functions such as the Boolean PVR, and that relates to the fact that certain networks tend to greedily learn monomials by incrementing their degree. In particular, this allows to characterize the generalization error in terms of the Boolean influence, a second central notion in Boolean Fourier analysis. Boolean measures thus seem to have a role to play in understanding generalization when learning of 'reasoning' or 'logical' functions. There are now many directions to pursue such as: (1) extending the realm of architectures/models for which we can prove formally the Boolean influence tightness, (2) considering more general holdout or distribution shift models, (3) investigate how the picture changes when the bits/digits are given by MNIST images, (4) better understanding when the low-degree implicit bias is taking place or not, within and beyond PVR, since the Boolean influence is not always tight in our experiments (e.g., MLP-Mixers seem to have a worse performance than the Boolean influence on PVR; see also Appendix F for other functions), (5) investigating how to 'revert' the implicit bias towards low-degree when it is taking place, to compensate for the unseen data; this will require justifying and engineering why certain symmetries are favorable in the learned function.

## Acknowledgments and Disclosure of Funding

We thank Raphaël Berthier (EPFL) and Jan Hązła (EPFL) for useful discussions.

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
