# A  Proof of Theorem 1

The proof of Theorem 1 goes by two steps. As a first step, we connect the noise stability to another measure of complexity for classes of functions called cross predictability (CP). As a second step, we use the negative result from [AS20], that lower bounds the generalization error of learning a class of functions in terms of its cross predictability.

## A.1  From noise stability to cross predictability

We redefine here the cross predictability (CP), for completeness.

**Definition 6** (Cross Predictability [AS20])**.** *Let $\mathcal{X}$ be the input space and let $\mathcal{F}$ be a class of functions. Let $P_\mathcal{X}$ and $P_\mathcal{F}$ be two distributions supported on $\mathcal{X}$ and $\mathcal{F}$ respectively. Their cross-predictability is defined as*

$$\mathrm{CP}(P_\mathcal{F}, P_\mathcal{X}) = \mathbb{E}_{F, F' \sim P_\mathcal{F}}[\mathbb{E}_{X \sim P_\mathcal{X}}[F(X) \cdot F'(X)]^2]. \tag{11}$$

Before diving into the proof we give few definitions that will be useful. Given a target function $f$, we define the "orbit" of $f$ ($\mathrm{orb}(f)$) as the class of all functions generated by composing $f$ with a permutation of the input space:

**Definition 7** (Orbit)**.** *For $f : \mathbb{R}^n \to \mathbb{R}$ and a permutation $\pi \in S_n$, we let $(f \circ \pi)(x) = f(x_{\pi(1)}, ..., x_{\pi(n)})$. Then, the* orbit *of $f$ is defined as*

$$\mathrm{orb}(f) := \{f \circ \pi : \pi \in S_n\}. \tag{12}$$

GD (or SGD) on a neural network with initialization that is target agnostic has equivalent behaviour when learning any target function in $\mathrm{orb}(f)$. We believe that one could extend the result to other invariances, beyond permutations.

Recall from Definition 5 that we introduced an augmented input space, to guarantee that the high-degree Fourier coefficients of the target function are sparse enough. Thus, let $\bar{f} : \{\pm 1\}^{2n} \to \{\pm 1\}$ be the $2n-$extension of $f$, defined as $\bar{f}(x_1, ..., x_n, x_{n+1}, ..., x_{2n}) = f(x_1, ..., x_n)$. For brevity we make use of the following notation:

$$\mathrm{CP}(\mathrm{orb}(\bar{f})) := \mathrm{CP}(\mathcal{U}_{\mathrm{orb}(\bar{f})}, \mathcal{U}_{\mathbb{F}_2^{2n}}), \tag{13}$$

where $\mathcal{U}_{\mathrm{orb}(\bar{f})}, \mathcal{U}_{\mathbb{F}_2^{2n}}$ denote the uniform distribution over $\mathrm{orb}(\bar{f})$ and over $\mathbb{F}_2^{2n}$ (i.e., the $2n$-dimensional Boolean hypercube), respectively.

Furthermore, recall that every Boolean function $f$ can be written in terms of its Fourier-Walsh expansion $f(x) = \sum_S \hat{f}(S)\chi_S(x)$, where $\chi_S(x) = \prod_{i \in S} x_i$ are the standard Fourier basis elements and $\hat{f}(S)$ are the Fourier coefficients of $f$. We further denote by

$$W^k(f) = \sum_{S:|S|=k} \hat{f}(S)^2 \quad \text{and} \quad W^{\leq k}(f) = \sum_{S:|S| \leq k} \hat{f}(S)^2, \tag{14}$$

the total weight of the Fourier coefficients of $f$ at degree $k$ and up to degree $k$, respectively.

Let $\hat{f}$ be the Fourier coefficients of the original function $f$, and let $\hat{h}$ be the coefficients of the augmented function $\bar{f}$, that are:

$$\hat{h}(T) = \hat{f}(T) \qquad \text{if } T \subseteq [n] \tag{15}$$

$$\hat{h}(T) = 0 \qquad \text{otherwise.} \tag{16}$$

We make use of the following Lemma, that relates the cross-predictability of $\mathrm{orb}(\bar{f})$ to the Stability of $f$.

**Lemma 2.** *There exists $\delta$ such that for any $\delta' < \delta$*

$$\mathrm{CP}(\mathrm{orb}(\bar{f})) \leq \mathrm{Stab}_{\delta'}(f). \tag{17}$$

*Remark* 4 (Noise Stability)**.**  We remark the following two properties of $\mathrm{Stab}_\delta[f]$:

1. One can show (see e.g., Theorem 2.49 in [O'D14]) that

$$\text{Stab}_\delta(f) = \sum_{k=1}^{n} (1 - 2\delta)^k W^k[f],\tag{18}$$

where $W^k[f]$ is the Boolean weight at degree $k$ of $f$;

2. For all $\delta \in [0, 1/2]$, $\text{Stab}_\delta(f) = \text{Stab}_\delta(\bar{f})$. This follows directly from the previous point and (15)-(16).

*Proof of Lemma 2.* We denote by $\pi$ a random permutation of $2n$ elements. We can bound the $\text{CP}(\text{orb}(\bar{f}))$ by the following:

$$\text{CP}(\text{orb}(\bar{f})) = \mathbb{E}_\pi \left[ \mathbb{E}_X \left[ \bar{f}(X) \bar{f}(\pi(X)) \right]^2 \right]\tag{19}$$

$$= \mathbb{E}_\pi \left( \sum_{T \subseteq [2n]} \hat{h}(T) \hat{h}(\pi(T)) \right)^2\tag{20}$$

$$= \mathbb{E}_\pi \left( \sum_{T \subseteq [n]} \hat{f}(T) \hat{f}(\pi(T)) \cdot \mathbb{1}\left(\pi(T) \subseteq [n]\right) \right)^2\tag{21}$$

$$\overset{C.S}{\leq} \mathbb{E}_\pi \left( \sum_{S \subseteq [n]} \hat{f}(\pi(S))^2 \right) \cdot \left( \sum_{T \subseteq [n]} \hat{f}(T)^2 \mathbb{1}\left(\pi(T) \subseteq [n]\right) \right)\tag{22}$$

$$= \sum_{T \subseteq [n]} \hat{f}(T)^2 \cdot \mathbb{P}_\pi \left(\pi(T) \subseteq [n]\right)\tag{23}$$

$$= \sum_{k=1}^{n} W^k[f] \cdot \mathbb{P}_\pi \left(\pi(T) \subseteq [n] \mid |T| = k\right),\tag{24}$$

where (20) is the scalar product in the Fourier basis, (21) follows by applying the formulas of the $\hat{h}$ given in (15)-(16), (22) holds by Cauchy-Schwarz inequality, (23) holds since $f$ is Boolean-valued and for each $\pi$ by Parseval identity, $\sum_{S \subseteq [n]} \hat{f}(\pi(S))^2 = \mathbb{E}_X[f(X)^2] = 1$, and (24) holds since the second term is invariant for all sets of a given cardinality.

Recalling $\pi$ is a random permutation over the augmented input space of dimension $2n$, for each $k \in [n]$ we can further bound the second term by

$$\mathbb{P}_\pi \left(\pi(T) \subseteq [n] \mid |T| = k\right) = \frac{\binom{n}{k}}{\binom{2n}{k}} \sim \frac{1}{2^k} \leq (1 - 2\delta')^k, \quad \text{for all } \delta' \leq 1/4.\tag{25}$$

Thus, for all $\delta' \leq 1/4$,

$$\text{CP}(\text{orb}(\bar{f})) \leq \sum_{k=1}^{n} (1 - 2\delta')^k W^k[f] = \text{Stab}_{\delta'}[f].\tag{26}$$

$\square$

*Remark* 5. Note that the value of $\delta$ in Lemma 2 depends on the size of the input extension that we use. In this paper, we defined an input extension of size $2n$ (input doubling), which gives $\delta = 1/4$, however we could have chosen e.g. a $3n$-extension and obtain $\delta = 1/3$, and so on.

## A.2 From cross predictability to hardness of learning

For the second step, we make use of Theorem 3 and Corollary 1 in [AS20], that prove a lower bound of learning a class of function in terms of its cross predictability. The lower bound holds for the noisy GD algorithm ([AS20, AKM$^+$21]), of which we give a formal definition here.

**Definition 8** (Noisy GD with batches). *Consider a neural network of size $E$, with a differentiable non-linearity and initialization of the weights $W^{(0)}$. Given a differentiable loss function, the updates of the noisy GD algorithm with learning rate $\gamma_t$ and gradient precision $A$ are defined by*

$$W^{(t)} = W^{(t-1)} - \gamma_t \mathbb{E}_{X \sim S^{(t)}}[\nabla L(f(X), f_{NN}^{(t)})]_A + Z^{(t)}, \qquad t = 1, ..., T, \qquad (27)$$

*where for all $t$, $Z^{(t)}$ are i.i.d. $\mathcal{N}(0, \sigma^2)$, for some $\sigma$, and they are independent from other variables, $S^{(t)} = (X_1^{(t)}, ..., X_m^{(t)})$ has independent components drawn from the input distribution $P_{\mathcal{X}}$ and independent from other time steps, and $f$ is the target function, from which the labels are generated, and by $[.]_A$ we mean that whenever the argument is exceeding $A$ (resp. $-A$) it is rounded to $A$ (resp. $-A$).*

Theorem 3 and Corollary 1 in [AS20] imply that for any distribution over the Boolean hypercube $P_{\mathcal{X}}$ and Boolean functions $P_{\mathcal{F}}$, it holds that

$$\mathbb{P}_{X, F \sim P_{\mathcal{F}}, f_{NN}^{(T)}}(F(X) \neq f_{NN}^{(T)}(X)) \geq 1/2 - \frac{\gamma T \sqrt{E} A}{\sigma} \left(1/m + \mathrm{CP}(P_{\mathcal{F}}, P_{\mathcal{X}})\right)^{1/4}, \qquad (28)$$

where $\gamma, E, A, \sigma, m$ have the same meaning as in Definition 8. As observed by them, in our case since the initialization is invariant under permutations of the input, then learning the orbit of $\bar{f}$ under uniform distribution is equivalent to learning $\bar{f}$, thus the following bound holds:

$$\mathbb{P}_{X, f_{NN}^{(T)}}(\bar{f}(X) \neq f_{NN}^{(T)}(X)) \geq 1/2 - \frac{\gamma T \sqrt{E} A}{\sigma} \left(1/m + \mathrm{CP}(\mathrm{orb}(\bar{f}))\right)^{1/4}. \qquad (29)$$

## B   Removing the input doubling

One can prove a similar result to the one of Theorem 1, without using the input extension technique. However, we need some additional assumptions on $f$. To introduce them, let us first fix some notation. In the following, we say that a sequence $a_n$ is *noticeable* if there exists $c \in \mathbb{N}$ such that $a_n = \Omega(n^{-c})$. On the other hand, we say that $f$ is *negligible* if $\lim_{n \to \infty} n^c a_n = 0$ for every $c \in \mathbb{N}$ (which we also write $a_n = n^{-\omega(1)}$).

**Assumption 1** (Non-dense and non-extremal function).

a)  *We say that $f$ is "non-dense" if there exists $c$ such that $W\{T : \hat{f}(T)^2 \leq n^{-c}\} = n^{-\omega(1)}$, i.e., the negligible Fourier coefficients do not bring a noticeable contribution if taken all together;*

b)  *We say $f$ is "non-extremal" if for any positive constant $D$, $W^{\geq n-D}[f] = n^{-\omega(1)}$, i.e., $f$ does not have noticeable Fourier weight on terms of degree $n - O(1)$.*

With such additional assumptions, we can conclude the following.

**Proposition 1.** *Let $f : \{\pm 1\}^n \to \{\pm 1\}$ be a balanced target function, let $\mathrm{Stab}_\delta(f)$ be its noise stability and let $f_{NN}^{(t)}$ be the output of $GD$ with gradient precision $A$ after $t$ time steps, trained on a neural network of size $E$ with initialization that is target agnostic. Assume $f$ satisfies Assumption 1. Then, there exist $c, C > 0$ and $D > 0$ such that if $\delta < D/n$*

$$\mathrm{gen}(f, f_{NN}^{(t)}) \geq 1/2 - C \cdot t \cdot \sqrt{E} \cdot \left(n^c \cdot \mathrm{Stab}_\delta(f) + n^{-\omega(1)}\right)^{1/4}. \qquad (30)$$

The proof of Proposition 1 resembles the proof of Theorem 1. The only modification required is in Lemma 2, which is replaced by the following Lemma.

**Lemma 3.** *Let $f$ be a Boolean function that satisfies Assumption 1. There exists $c, D > 0$ such that for $\delta < D/n$,*

$$\mathrm{CP}(\mathrm{orb}(f)) \leq 2 \cdot n^c \cdot \mathrm{Stab}_\delta(f) + n^{-\omega(1)}. \qquad (31)$$

*Proof of Lemma 3.* Let $c > 0$ be such that $W\{T : \hat{f}(T)^2 \leq n^{-c}\} = n^{-\omega(1)}$. This $c$ exists because of Assumption 1a.

$$\text{CP}(\text{orb}(f)) = \tag{32}$$

$$= \mathbb{E}_\pi \left[ \mathbb{E}_X \left[ f(X) f(\pi(X)) \right]^2 \right] \tag{33}$$

$$= \mathbb{E}_\pi \left( \sum_{T \subseteq [n]} \hat{f}(T) \hat{f}(\pi(T)) \right)^2 \tag{34}$$

$$= \mathbb{E}_\pi \left( \sum_{T \subseteq [n]} \hat{f}(T) \hat{f}(\pi(T)) \cdot \left( \mathbb{1}\left( \hat{f}(\pi(T))^2 \leq n^{-c} \right) + \mathbb{1}\left( \hat{f}(\pi(T))^2 > n^{-c} \right) \right) \right)^2 \tag{35}$$

$$\leq 2\mathbb{E}_\pi \left( \sum_{T \subseteq [n]} \hat{f}(T) \hat{f}(\pi(T)) \mathbb{1}\left( \hat{f}(\pi(T))^2 \leq n^{-c} \right) \right)^2 + \tag{36}$$

$$+ 2\mathbb{E}_\pi \left( \sum_{T \subseteq [n]} \hat{f}(T) \hat{f}(\pi(T)) \mathbb{1}\left( \hat{f}(\pi(T))^2 > n^{-c} \right) \right)^2, \tag{37}$$

where in the last inequality we used $(a + b)^2 \leq 2(a^2 + b^2)$. Let us first focus on the second term on the right.

$$\mathbb{E}_\pi \left( \sum_{T \subseteq [n]} \hat{f}(T) \hat{f}(\pi(T)) \mathbb{1}\left( \hat{f}(\pi(T))^2 > n^{-c} \right) \right)^2 \tag{38}$$

$$\overset{C.S}{\leq} \mathbb{E}_\pi \left( \sum_{S \subseteq [n]} \hat{f}(\pi(S))^2 \right) \cdot \left( \sum_{T \subseteq [n]} \hat{f}(T)^2 \mathbb{1}\left( \hat{f}(\pi(T))^2 > n^{-c} \right) \right) \tag{39}$$

$$\leq \sum_{T \subseteq [n]} \hat{f}(T)^2 \cdot \mathbb{P}_\pi \left( \hat{f}(\pi(T))^2 > n^{-c} \right) \tag{40}$$

$$= \sum_{k=1}^{n} W^k[f] \cdot \mathbb{P}_\pi \left( \hat{f}(\pi(T))^2 > n^{-c} \mid |T| = k \right) \tag{41}$$

$$= \sum_{k=1}^{n-D} W^k[f] \cdot \mathbb{P}_\pi \left( \hat{f}(\pi(T))^2 > n^{-c} \mid |T| = k \right) + \tag{42}$$

$$+ \sum_{k=n-D+1}^{n} W^k[f] \cdot \mathbb{P}_\pi \left( \hat{f}(\pi(T))^2 > n^{-c} \mid |T| = k \right)$$

$$\leq \sum_{k=1}^{n-D} W^k[f] \cdot \mathbb{P}_\pi \left( \hat{f}(\pi(T))^2 > n^{-c} \mid |T| = k \right) + W^{\geq n-D+1}[f]. \tag{43}$$

where $D$ is an arbitrary positive constant. Because of Assumption 1b, $W^{\geq n-D+1}[f] = n^{-\omega(1)}$. On the other hand, since $f$ is a Boolean valued function,

$$\sum_{T} \hat{f}(T)^2 = \mathbb{E}_X[f(X)^2] = 1, \tag{44}$$

which implies that there are at most $n^c$ sets $T$ such that $\hat{f}(T)^2 > n^{-c}$. Thus, recalling $\pi$ is a random permutation over the input space of dimension $n$, we get

$$\mathbb{P}_\pi \left( \hat{f}(\pi(T))^2 > n^{-c} \mid |T| = k \right) \leq \frac{n^c}{\binom{n}{k}} \tag{45}$$

$$\leq n^c \left( \frac{k}{n} \right)^k \tag{46}$$

$$\leq n^c \left( \frac{n-D}{n} \right)^k \tag{47}$$

$$\leq n^c \left( 1 - 2\delta \right)^k \qquad \text{if } \delta \leq \frac{D}{2n}, \tag{48}$$

where in (46) we used that $\binom{n}{k} \geq (\frac{n}{k})^k$ for all $k \geq 1$. Going back to the first term in (37) we get

$$\mathbb{E}_\pi \left( \sum_{T \subseteq [n]} \hat{f}(T)\hat{f}(\pi(T)) \mathbb{1} \left( \hat{f}(\pi(T))^2 \leq n^{-c} \right) \right)^2 \tag{49}$$

$$\overset{C.S}{\leq} \mathbb{E}_\pi \left( \sum_{S \subseteq [n]} \hat{f}(S)^2 \right) \cdot \left( \sum_{T \subseteq [n]} \hat{f}(\pi(T))^2 \mathbb{1} \left( \hat{f}(\pi(T))^2 > n^{-c} \right) \right) \tag{50}$$

$$\leq \sum_{T \subseteq [n]} \hat{f}(\pi(T))^2 \mathbb{1} \left( \hat{f}(\pi(T))^2 > n^{-c} \right) \tag{51}$$

$$= n^{-\omega(1)}, \tag{52}$$

by Assumption 1a. Hence overall,

$$\mathrm{CP}(\mathrm{orb}(f)) \leq 2n^c \sum_{k=1}^{n-D} W^k[f](1-2\delta)^k + n^{-\omega(1)} \tag{53}$$

$$\leq 2n^c \, \mathrm{Stab}_\delta(f) + n^{-\omega(1)}. \tag{54}$$

$\square$

## C  Proof for Lemma 1 and Theorem 2

In this section, we present proofs for results mentioned in Section 3, namely, Lemma 1 and Theorem 2.

### C.1  Proof of Lemma 1

*Proof of Lemma 1.* Let $f(x) = \sum_{T \subseteq [n]} \hat{f}(T)\chi_T(x)$ be the Fourier expansion of the function where $\chi_T(x) = \prod_{i \in T} x_i$. Therefore, the Fourier expansion of the frozen function will become

$$f_{-k}(x) = \sum_{T \subseteq [n] \setminus k} (\hat{f}(T) + \hat{f}(T \cup k))\chi_T(x). \tag{55}$$

Thus, the difference between functions is equal to

$$(f - f_{-k})(x) = \sum_{T \subseteq [n]: k \in T} \hat{f}(T)\chi_T(x) - \sum_{T \subseteq [n] \setminus k} \hat{f}(T \cup k)\chi_T(x). \tag{56}$$

Hence, using Parseval's Theorem we have the following:

$$\mathbb{E}_{U^n}(f - f_{-k})_2^2 = \sum_{T \subseteq [n]: k \in T} \hat{f}(T)^2 + \sum_{T \subseteq [n] \setminus k} \hat{f}(T \cup k)^2 = 2 \sum_{T \subseteq [n]: k \in T} \hat{f}(T)^2. \tag{57}$$

Therefore,

$$\mathbb{E}_{U^n} \frac{1}{2}(f - f_{-k})_2^2 = \sum_{T \subseteq [n]: k \in T} \hat{f}(T)^2 = \mathrm{Inf}_k(f), \tag{58}$$

and the lemma is proved. $\square$

## C.2 Proof of Theorem 2

*Proof of Theorem 2.* Assume $\tilde{f}_{-k}^{(t)}(x, \Theta^{(t)}) := x^T W^{(t)} + b^{(t)}$ to be our linear model where $\Theta^{(t)} = (W^{(t)}, b^{(t)})$ are the model parameters at time $t$. In the following, the super-script $t$ and $T$ denote the time-step and transpose respectively. Also, we use $\mathbb{E}_{x_{-k}}$ to denote the expectation of $x$ taken uniformly on the Boolean hypercube while $x_k = 1$. Using the square loss, we have

$$L(\Theta^{(t)}, x, f) = (x^T W^{(t)} + b^{(t)} - f(x))^2, \tag{59}$$

and the gradients will be

$$\nabla_W L(\Theta^{(t)}, x, f) = 2x \left( x^T W^{(t)} + b^{(t)} - f(x) \right), \tag{60}$$

$$\partial_b L(\Theta^{(t)}, x, f) = 2 \left( x^T W^{(t)} + b^{(t)} - f(x) \right). \tag{61}$$

The GD update rule will then become

$$W^{(t+1)} = W^{(t)} - 2\gamma \left( \mathbb{E}_{x_{-k}} \left[ xx^T \right] W^{(t)} + \mathbb{E}_{x_{-k}}[x] b^{(t)} - \mathbb{E}_{x_{-k}}[xf(x)] \right), \tag{62}$$

$$b^{(t+1)} = b^{(t)} - 2\gamma \left( \mathbb{E}_{x_{-k}}[x^T] W^{(t)} + b^{(t)} - \mathbb{E}_{x_{-k}}[f(x)] \right). \tag{63}$$

Note that $\mathbb{E}_{x_{-k}} \left[ xx^T \right] = \mathbb{I}_n, \mathbb{E}_{x_{-k}}[x] = \vec{e}_k$. So we have

$$\forall j \neq k: \ W_j^{(t+1)} = W_j^{(t)}(1 - 2\gamma) + 2\gamma \mathbb{E}_{x_{-k}}[x_j \cdot f(x)], \tag{64}$$

$$W_k^{(t+1)} = W_k^{(t)} - 2\gamma(W_k^{(t)} + b^{(t)}) + 2\gamma \mathbb{E}_{x_{-k}}[f(x)], \tag{65}$$

$$b^{(t+1)} = b^{(t)} - 2\gamma(W_k^{(t)} + b^{(t)}) + 2\gamma \mathbb{E}_{x_{-k}}[f(x)]. \tag{66}$$

Using above equations, we have

$$W_k^{(t+1)} - b^{(t+1)} = W_k^{(t)} - b^{(t)} = W_k^{(0)} - b^{(0)}, \tag{67}$$

$$W_k^{(t+1)} + b^{(t+1)} = (1 - 4\gamma)(W_k^{(t)} + b^{(t)}) + 4\gamma \mathbb{E}_{x_{-k}}[f(x)]. \tag{68}$$

Assume $\gamma < \frac{1}{4}$ and define $0 < c = -\log(1 - 2\gamma) < -\log(1 - 4\gamma)$, then we have

$$\begin{aligned} W_k^{(t)} + b^{(t)} &= (1 - 4\gamma)^t (W_k^{(0)} + b^{(0)} - \mathbb{E}_{x_{-k}}[f(x)]) + \mathbb{E}_{x_{-k}}[f(x)] \\ &= O((1 - 4\gamma)^t) + \mathbb{E}_{x_{-k}}[f(x)] = O(e^{-ct}) + \mathbb{E}_{x_{-k}}[f(x)] \\ &= O(e^{-ct}) + \hat{f}(\emptyset) + \hat{f}(\{k\}), \end{aligned} \tag{69}$$

$$\begin{aligned} \forall j \neq k: \ W_j^{(t)} &= (1 - 2\gamma)^t (W_j^{(0)} - \mathbb{E}_{x_{-k}}[x_j \cdot f(x)]) + \mathbb{E}_{x_{-k}}[x_j \cdot f(x)] \\ &= O((1 - 2\gamma)^t) + \mathbb{E}_{x_{-k}}[x_j \cdot f(x)] = O(e^{-ct}) + \mathbb{E}_{x_{-k}}[x_j \cdot f(x)] \\ &= O(e^{-ct}) + \hat{f}(\{j\}). \end{aligned} \tag{70}$$

So the learned function is

$$\tilde{f}_{-k}(x; \Theta^{(t)}) = \frac{b^{(0)} - W_k^{(0)} + \hat{f}(\emptyset) + \hat{f}(\{k\})}{2} + \frac{W_k^{(0)} - b^{(0)} + \hat{f}(\emptyset) + \hat{f}(\{k\})}{2} x_k$$
$$+ \sum_{j \neq k} \hat{f}(\{j\}) \cdot x_j + O(e^{-ct}) \tag{71}$$

and the generalization error can be computed using Parseval Theorem:

$$\text{gen}(f, \tilde{f}_{-k}^{(t)}) = \frac{1}{2} \mathbb{E}_{x \sim U^n} \left[ \left( f(x) - \tilde{f}_{-k}^{(t)}(x; \Theta^\infty) \right)^2 \right] \tag{72}$$

$$= \frac{1}{2} \left( \frac{(b^{(0)} - W_k^{(0)} - \hat{f}(\emptyset) + \hat{f}(\{k\}))^2 + (W_k^{(0)} - b^{(0)} + \hat{f}(\emptyset) - \hat{f}(\{k\}))^2}{4} \right) + O(e^{-ct}) \tag{73}$$

$$= \frac{(b^{(0)} - W_k^{(0)} - \hat{f}(\emptyset) + \hat{f}(\{k\}))^2}{4} + O(e^{-ct}) \tag{74}$$

$$= \frac{(b^{(0)} - W_k^{(0)})^2}{4} + \frac{(\hat{f}(\emptyset) - \hat{f}(\{k\}))^2}{4} - 2\frac{(b^{(0)} - W_k^{(0)})(\hat{f}(\emptyset) - \hat{f}(\{k\}))}{4} + O(e^{-ct}). \tag{75}$$

Therefore, the expected generalization loss over different initializations is given by

$$\mathbb{E}_{\Theta^0}[\text{gen}(f, \tilde{f}_{-k}^{(t)})] = \mathbb{E}_{\Theta^0}\left[\frac{(b^{(0)} - W_k^{(0)})^2 + (\hat{f}(\emptyset) - \hat{f}(\{k\}))^2}{4}\right] + O(e^{-ct}) \qquad (76)$$

$$= \frac{(\hat{f}(\emptyset) - \hat{f}(\{k\}))^2}{4} + \frac{\sigma^2}{2} + O(e^{-ct}). \qquad (77)$$

Particularly, if the frozen function is unbiased, i.e., $\hat{f}(\emptyset) + \hat{f}(\{k\}) = 0$, we have

$$\mathbb{E}_{\Theta^0}[\text{gen}(f, \tilde{f}_{-k}^{(t)})] = \frac{(2\hat{f}(\{k\}))^2}{4} + \frac{\sigma^2}{2} + O(e^{-ct})$$

$$= \hat{f}(\{k\})^2 + \frac{\sigma^2}{2} + O(e^{-ct}) = \text{Inf}_k(f) + \frac{\sigma^2}{2} + O(e^{-ct}). \qquad (78)$$

$\square$

## D  Further details on noise stability

### D.1  Noise stability of PVR functions

As mentioned above, a PVR function consists of a pointer (the first bits of the input) and an aggregation function that acts on a specific window indicated by the pointer. We denote by $p$ the number of bits that define the pointer, and by $w$ the size of each window. For simplicity, we consider a slight variation of Boolean PVR task with non-overlapping windows, defined as follows:

- PVR with *non-overlapping* windows: the $2^p$ windows pointed by the pointer bits are non-overlapping, i.e., the first window is formed by bits $x_{p+1}, ..., x_{p+w}$, the second window is formed by bits $x_{p+w+1}, ..., x_{p+2w}$, and so forth.

The input size is thus given by $n := p + 2^p w$ and $p = O(\log(n))$. We denote by $g : \{\pm 1\}^w \to \{\pm 1\}$ the aggregation function, which we assume to be balanced (i.e., $\mathbb{E}_X[g(X)] = 0$). One can verify (see details below) that the noise stability of the PVR function $f$ is given by

$$\text{Stab}_\delta[f] = (1 - \delta)^{p+w} + (1 - \delta)^p (1 - (1 - \delta)^w) \cdot \text{Stab}_\delta[g]. \qquad (79)$$

We notice that the $\text{Stab}_\delta[f]$ is given by two terms: the first one depends on the window size and the second one on the stability of the aggregation function. For large enough window size, the second term in (79) is the dominant one, and $\text{Stab}_\delta[f]$ depends on the stability of $g$. Thus from Theorem 1, $f$ is not learned by GD (in the extended input space) in poly(n) time if the stability of the aggregation function is $n^{-\omega(1)}$. On the other hand, for small window size (specifically for $w = O(\log(n))$), the $\text{Stab}_\delta(f)$ is 'noticeable' (as defined in Appendix B) for every aggregation function, since the function value itself depends on a limited number of input bits. Thus, noise unstable aggregation functions (e.g. parities) can form a PVR function with 'noticeable' stability, if the window size is $O(\log(n))$. As examples, we consider the specific cases of pairty and majority vote as aggregation functions.

- Parity: If we choose $g(x_1, ..., x_w) = \prod_{i=1}^w x_i$, one can observe that $\text{Stab}_\delta(g) = (1 - 2\delta)^w$. Then, eq. (79) becomes $\text{Stab}_\delta(f) = (1 - \delta)^{w+p}[1 - (1 - 2\delta)^w] + (1 - \delta)^p$, and $\text{Stab}_\delta(f)$ is decreasing with $w$.
- Majority: If we choose $g$ to be $g(x_1, ..., x_w) = \text{sgn}(\sum_{i=1}^w x_i)$, then, for $w$ large, $\text{Stab}_\delta(g) \sim 1 - 2/\pi \cdot \arccos(1 - 2\delta)$ (see e.g. [O'D14]). Plugging this in eq. (79), one can observe that also for majority vote $\text{Stab}_\delta(f)$ is decreasing with $w$.

**Computation of** (79). We compute the expression in (79) with the following:

$$\text{Stab}_\delta[f] = 1 - 2\,\text{NS}_\delta[f], \qquad (80)$$

where $\text{NS}_\delta[f] := \mathbb{P}(f(X) \neq f(Y))$ is the Noise sensitivity of $f$, defined as the probability that perturbing each input bit independently with probability $\delta$ changes the output of $f$ and where we denoted by $Y$ the vector obtained from $X$ by flipping each component with prob. $\delta$ independently. To

compute $\mathrm{NS}_\delta[f]$, we can first distinguish depending on whether the perturbation affects the pointer bit:

$$\mathrm{NS}_\delta[f] := \mathbb{P}(f(X) \neq f(Y))$$
$$= (1-\delta)^p \cdot \mathbb{P}(f(X) \neq f(Y) \mid X^p = Y^p) + (1-(1-\delta)^p)\mathbb{P}(f(X) \neq f(Y) \mid X^p \neq Y^p)$$
$$= (1-\delta)^p \cdot \mathbb{P}(f(X) \neq f(Y) \mid X^p = Y^p) + (1-(1-\delta)^p)\frac{1}{2},$$

where the last inequality holds since we are using non-overlapping windows and we assumed $g$ to be balanced. To compute the first term, we can condition on whether any bit in the window pointed by $X$ and $Y$ is changed:

$$\mathbb{P}(f(X) \neq f(Y) \mid X^p = Y^p)$$
$$= (1-\delta)^w \cdot \mathbb{P}(f(X) \neq f(Y) \mid X^p = Y^p, X_{P(X^p)} = Y_{P(Y^p)}) +$$
$$+ (1-(1-\delta)^w) \cdot \mathbb{P}(f(X) \neq f(Y) \mid X^p = Y^p, X_{P(X^p)} \neq Y_{P(Y^p)})$$
$$= (1-(1-\delta)^w) \cdot \mathbb{P}(f(X) \neq f(Y) \mid X^p = Y^p, X_{P(X^p)} \neq Y_{P(Y^p)})$$
$$= (1-(1-\delta)^w) \cdot \mathrm{NS}_\delta[g],$$

where the last inequality holds because $g$ is unbalanced. By replacing $\mathrm{NS}_\delta[g] = \frac{1}{2} - \frac{1}{2}\mathrm{Stab}_\delta[g]$ and rearranging terms one can obtain (79).

### D.2   Noise stability and initial alignment [ACHM22]

[ACHM22] introduced the notion of Initial Alignment (INAL) between a target function $f : \mathcal{X} \to \mathcal{Y}$ and a neural network $\mathrm{NN} : \mathcal{X} \to \mathcal{Y}$ with random initialization $\Theta_0$ and neuron set $V_{NN}$. The INAL is defined as

$$\mathrm{INAL}(f,\mathrm{NN}) := \max_{v \in V_{\mathrm{NN}}} \mathbb{E}_{\Theta^0}\mathbb{E}_X \left[ f(X) \cdot \mathrm{NN}_{\Theta^0}^{(v)}(X) \right]^2, \tag{81}$$

where $\mathrm{NN}_{\Theta^0}^{(v)}$ denotes the output of neuron $v$ of the network at initialization. In [ACHM22], it is shown that GD cannot learn functions that have negligible initial alignment with a fully connected architectures with i.i.d. Gaussian initialization (with rescaled variance) and ReLU activation. Here, we show how the INAL can be related to the noise sensitivity of the target function. We remark that both noise sensitivity and INAL are related by the cross-predictability (CP). Let us first give two definitions. Recall that for $f : \{\pm 1\}^n \to \{\pm 1\}$, $\mathrm{NS}_\delta[f] = \frac{1}{2} - \frac{1}{2}\mathrm{Stab}_\delta[f]$.

**Definition 9** (High-Degree.). *We say that a family of functions $f_n : \{\pm 1\}^n \to \mathbb{R}$ is "high-degree" if for any fixed $k$, $W^{\leq k}(f_n)$ is negligible.*

**Definition 10** (Noise sensitive function). *We say that a family of functions $f_n : \{\pm 1\}^n \to \{\pm 1\}$ is noise sensitive if for any $\delta \in (0,1/2]$, $\mathrm{NS}_\delta[f_n] = 1/2 - o_n(1)$.*

**Definition 11** (Strongly noise sensitive function). *We say that a family on functions $f_n : \{\pm 1\}^n \to \{\pm 1\}$ is strongly noise sensitive if for any $\delta \in (0,1/2]$, $\mathrm{NS}_\delta[f_n] = 1/2 - n^{-\omega(1)}$.*

Then we can prove the following.

**Proposition 2.** *Let $\mathrm{NN}_n : \mathbb{R}^n \to \mathbb{R}$ be a fully connected neural network with Gaussian i.i.d. initialization and expressive activation (as in Theorem 2.7 in [ACHM22]). If $\mathrm{INAL}(\mathrm{NN}_n, f_n) = n^{-\omega(1)}$, then $f_n$ is noise sensitive.*

*Proof.* We need to show that for any $\delta \in [0,1/2]$, $\sum_{k=0}^{n}(1-2\delta)^k W^k(f_n) = o_n(1)$, or analogously that for any $\epsilon > 0$ and for $n$ large enough $\sum_{k=0}^{n}(1-2\delta)^k W^k(f_n) < \epsilon$. Fix $\delta$ and let $\epsilon > 0$. Let $k_0$ be such that $(1-2\delta)^{k_0} < \epsilon/2$. Then,

$$\sum_{k=0}^{n}(1-2\delta)^k W^k(f_n) = \sum_{k=0}^{k_0}(1-2\delta)^k W^k(f_n) + \sum_{k=k_0+1}^{n}(1-2\delta)^k W^k(f_n) \tag{82}$$

$$\leq W^{\leq k_0}(f_n) + (1-2\delta)^{k_0+1}\sum_{k=k_0+1}^{n} W^k(f_n). \tag{83}$$

By Proposition 4.3 and Corollary 4.4 in [ACHM22], if $\mathrm{INAL}(f_n, \sigma) = n^{-\omega(1)}$ then $f_n$ is high degree. Thus, $W^{\leq k_0}(f_n) = n^{-\omega(1)}$, and clearly for $n$ large enough $W^{\leq k_0}(f_n) < \epsilon/2$. On the other hand, $\sum_{k=k_0+1}^{n} W^k(f_n) < 1$, since $f$ is Boolean-valued. Thus, $\sum_{k=0}^{n}(1 - 2\delta)^k W^k(f_n) < \epsilon$, and the Proposition is proven. $\qquad\square$

**Proposition 3.** *If $f_n$ is strongly noise sensitive, then $f_n$ is high degree.*

*Proof.* We need to show that if $\sum_{k=0}^{n}(1 - 2\delta)^k W^k(f_n) = n^{-\omega(1)}$ then for any constant $k$, $W^{\leq k}(f_n) = n^{-\omega(1)}$. Take $k_0 \in \mathbb{N}$, then

$$n^{-\omega(1)} = \sum_{k=0}^{n}(1 - 2\delta)^k W^k(f_n) \geq \sum_{k=0}^{k_0}(1 - 2\delta)^k W^k(f_n) \geq (1 - 2\delta)^{k_0} W^{\leq k_0}(f_n). \tag{84}$$

Clearly this implies that $W^{\leq k_0}(f_n) = n^{-\omega(1)}$, and the proof is concluded. $\qquad\square$

# E Computation of the Boolean influence for PVR functions

In this section, we compute the Boolean influence for PVR functions. Here, we consider PVR functions with sliding windows and cyclic indices (i.e., $x_{n+1} = x_{p+1}$). The Boolean influence for PVR tasks with truncated windows or non-overlapping windows can be calculated in a similar manner. Also note that we never freeze pointer bits in this paper as done in [ZRKB21]; therefore, we skip the calculation of the Boolean influence for pointer bits. Consider a bit at $k$-th position ($k > p$). Note that this bit appears in $w$ windows. We denote by $U^n$ the uniform distribution over the $n$-dimensional hypercube. Using Lemma 1, we have:

$$\mathrm{Inf}_k(f) = \mathbb{E}_{x \sim U^n} \frac{1}{2} \left( f(x) - f_{-k}(x) \right)^2 \tag{85}$$

$$= \mathbb{E}_{x \sim U^n} \frac{1}{2} \left( \sum_{i=0}^{w-1} \mathbb{1}(P(x^p) = k - i) \Big( g(x_{k-i}, \ldots, x_k, \ldots, x_{k-i+w-1}) \right. \tag{86}$$

$$\left. - g(x_{k-i}, \ldots, 1, \ldots, x_{k-i+w-1}) \Big) \right)^2$$

$$= \mathbb{E}_{x \sim U^n} \frac{1}{2} \left( \sum_{i=0}^{w-1} \mathbb{1}(P(x^p) = k - i) \Big( g(x_{k-i}, \ldots, x_k, \ldots, x_{k-i+w-1}) \right. \tag{87}$$

$$\left. - g(x_{k-i}, \ldots, 1, \ldots, x_{k-i+w-1}) \Big)^2 \right)$$

$$= \frac{1}{2^p} \sum_{i=1}^{w} \mathrm{Inf}_i(g). \tag{88}$$

Note that the expression $\sum_{i=1}^{w} \mathrm{Inf}_i(g)$ in Equation (88) is known as the *total influence* of the aggregation function $g$ [O'D14]. Below follows the value of the Boolean influence of the PVR task $f$, depending on different aggregation functions:

- **Parity.** If we choose $g$ to be the parity function, i.e., $g(x_1, \ldots, x_w) = x_1 x_2 \cdots x_w$ then $\mathrm{Inf}_i(g) = \mathbb{P}(g(x) \neq g(x + e_i)) = 1$. Therefore, $\mathrm{Inf}_k(f) = \frac{w}{2^p}$.
- **Median/Majority vote.** We define the majority vote function as $g(x_1, \ldots, x_w) = \mathrm{sign}(x_1 + \cdots + x_w)$ where the sign function outputs $+1$, $-1$, and $0$. First assume $w$ is odd. In this case, flipping the $i$-th bit matters only in the case where exactly $\frac{w-1}{2}$ other bits have the same sign as the $i$-th bit. Therefore, $\mathrm{Inf}_i(g) = \mathbb{P}(g(x) \neq g(x + e_i)) = 2^{-(w-1)} \binom{w-1}{\frac{w-1}{2}}$. Similarly, if $w$ is even, flipping the $i$-th bit only matters if there are exactly $\frac{w}{2}$ or $\frac{w}{2} - 1$ other bits with the same sign. Using Lemma 1, $\mathrm{Inf}_i(g) = \mathbb{E}_{x \sim U^w} \frac{1}{2}(g(x) - g_{-i}(x))^2 = 2^{-(w+1)} \left( \binom{w-1}{\frac{w}{2}} + \binom{w-1}{\frac{w}{2}-1} \right) = 2^{-w} \binom{w-1}{\frac{w}{2}}$. Therefore, for odd $w$, $\mathrm{Inf}_k(f) = \frac{w}{2^{(p+w-1)}} \binom{w-1}{\frac{w-1}{2}}$ and for even $w$, $\mathrm{Inf}_k(f) = \frac{w}{2^{(w+p)}} \binom{w-1}{\frac{w}{2}}$.

- **Min/Max.** Here we consider the min function, $g(x_1, \ldots, x_w) = \min(x_1, \ldots, x_w)$. By symmetry, the Boolean influence values are the same for the max function. In this case, flipping the $i$-th bit only matters if all bits other than $x_i$ are equal to $+1$. Thus, the Boolean influence is given by $\mathrm{Inf}_i(g) = 2^{-(w-1)}$ and hence, $\mathrm{Inf}_k(f) = \frac{w}{2^{(p+w-1)}}$.

One can see how different parameters of the Boolean PVR functions, such as $p$, $w$, and $g$ affect the Boolean influence. Assuming fixed window size, $w$, each bit is less likely to appear in a window if the number of pointer bits, $p$, is increased. Hence for fixed $w$ and $g$, an increase in $p$ results in smaller influence for all the bits. On the other hand a change of $w$ has a two-fold effect. First, since each bit appears in $w$ windows, the increase of $w$ makes each bit more likely to appear in a window. On the other hand, for some functions such as majority-vote and min/max, the increase of $w$ reduces the Boolean influence of the aggregation function for all bits. Thus, the increase of $w$ can result in either an increase of the Boolean influence (for example, if parity is used) or a decrease of the Boolean influence (for instance, if min/max aggregation is used). We refer to Appendix F for experiments on PVR tasks with varying window size.

# F    Experiment Details and Additional Experiments

In this section, we describe the experiments in more detail. Furthermore, we demonstrate more experiments on the comparison of the out-of-distribution generalization error and the Boolean influence.

## F.1    Architectures and Procedure

We first explain the general experimentation setup for PVR tasks and other functions. Afterward, we describe the procedure used for linear neural networks and results presented in Section 3.3.

**Architectures.**    Three architectures have been used for the main experiments of this paper: MLP, the Transformer [VSP$^+$17], and MLP-Mixer [THK$^+$21]. Below, we describe each of these architectures:

- **MLP.** The MLP model consists of 4 fully connected hidden layers of sizes 512, 1024, 512, and 64. We used ReLU as the activation function for all layers except the last layer.
- **Transformer.** We follow the standard decoder-only Transformer architectures [RSR$^+$19] that are commonly used for language modeling, and are also the backbone of Vision Transformers (ViTs) [DBK$^+$20]. Specifically, an embedding layer is used to embed the binary $+1$ and $-1$ values into 256 dimensional vectors, and a shared embedding layer is used for all the binary tokens in the input sequence. Then, the embedded input is passed through 12 transformer layers [VSP$^+$17]. In each transformer layer, the hidden dimension of MLP block is also 256. Moreover, 6 heads are used for each self-attention block. At the end, a linear layer is used to compute the output of the model.
- **MLP-Mixer.** Similar to the Transformer based model, first we embed $+1$ and $-1$ tokens into a 256 dimensional vector using a shared embedding layer for all the binary input tokens. Then, the embedded input is passed through a standard 12-layer MLP-Mixer model [THK$^+$21]. Finally, a linear layer is used to compute the output. The MLP-Mixer architectures are similar to the decoder-only Transformers, except that "mixer layers" based on MLPs are used instead of the attention mechanism. Please see [THK$^+$21] for details.

**Procedure.**    To perform each of the experiments, we first fix a dimension to be frozen during the training. Afterward, we train the model on the frozen training set to make the model learn the frozen function. Finally, we evaluate the trained model uniformly on the Boolean hypercube ($\{\pm 1\}^n$) to compute the out-of-distribution generalization error.

Now, we explain the hyperparameters used for the experiments. Note that the experiments are aimed to exhibit an implicit bias towards low degree monomials and consequently to show that the generalization error is close to the Boolean influence. Therefore, the experiments are not focused on the learning of the frozen function itself and the in-distribution generalization error. In other words, we are interested in the setting that the frozen function is learned during the training, and then we want to examine the out-of-distribution generalization. Due to this reason, we have always used a

relatively large number of training samples. Moreover, we have not used a learning rate scheduler and we have not done extensive hyperparameter tuning. Generally, we have considered two optimizers for the training of the models: mini-batch SGD and Adam [KB14]. We describe the setting used for each of them below:

- **SGD.** When using SGD, we tried using $0$ and $0.9$ momentum. We observed that the use of momentum remarkably accelerates the training process, and hence we continued with $0.9$ momentum. Nonetheless, we also performed experiments using SGD without momentum, and we did not notice any difference in their preference towards low degree monomials and therefore the out-of-distribution generalization error. For learning rate of SGD, we tried values in $\{10^{-3}, 5 \times 10^{-4}, 2.5 \times 10^{-4}, 10^{-4}\}$ and selected the learning rate dependent on the model and task (more on this below). Additionally, we always used the mini-batch size of 64 with SGD.

- **Adam.** In our experiments with Adam, we used default values of the optimizer and only changed the learning rate. For learning rate, we tried values in $\{10^{-4}, 5 \times 10^{-5}, 10^{-5}\}$ and finally selected $5 \times 10^{-4}$. While employing Adam, we used mini-batch size of 64 for PVR tasks with 3 pointer bits (11 bits in total) and mini-batch size of 1024 for PVR tasks with 4 pointer bits (20 bits in total).

We selected the learning rate (and in general, hyperparameters) based on the speed of the convergence and its stability. Note that we set the number of epochs for each task to a value to ensure the training loss and in-distribution generalization error are small enough.[10]

Finally, we note that all of our experiments are implemented using PyTorch framework [PGM+19], and the training has been done on NVIDIA A100 GPUs. The experiments presented in this paper took approximately 250 GPU hours. Note that we have repeated PVR experiments with 3 pointer bits 40 times and the rest of the experiments 20 times, and have reported the averaged results and $95\%$ confidence interval. Please refer to the code for more details on the experiments.

**Linear neural networks.** For the experiments on linear models, we considered fully connected linear neural networks with fixed hidden layer size of 256. As presented in Figure 4, we varied the initialization and depth of these networks. For optimizing linear neural networks, we used mini-batch SGD with 64 and $10^{-5}$ as the batch size and learning rate respectively. Note that we trained linear models on CPU and stopped the training when the loss became less than $10^{-8}$.

## F.2 Additional Results

**More PVR tasks.** First, we compare the generalization error and the Boolean influence for more PVR functions. In the additional experiments, we consider the cyclic version of the PVR (i.e., $x_{n+1} = x_{p+1}$), and due to the symmetry, we only freeze one dimension of the input. Also, we use Adam to optimize the models (instead of SGD) due to faster convergence. As a first example, we consider PVR tasks with 3 pointer bits (11 bits in total) and varying window sizes. We use majority-vote and parity as the aggregation functions (see Appendix E for computation of the Boolean influence for such functions). In Figure 5, the window size of the aforementioned PVR tasks is varied in the x-axis and the averaged generalization error over 40 experiments is shown. Figure 5 (top) corresponds to the case where majority-vote is used as the aggregation function whereas Figure 5 (bottom) shows the results when parity is the aggregation function. It can be seen that in this setting, the generalization error of all models follow the Boolean influence closely. Note that learning parity function becomes increasingly difficult as the window size is increased. Even for $w = 4$, the MLP and MLP-Mixer models could not learn the frozen function completely and their in-distribution generalization loss was between $0.05$ and $0.10$.

Furthermore, we experimented on PVR tasks of larger scales. To this end, we consider PVR tasks with 4 pointer bits (20 bits in total) and different window sizes and aggregation functions. For these experiments, we also used Adam optimizer with batch-size of 1024. We repeated each experiment 20 times. The generalization error and Boolean influence for these functions are given in Table 1. It can be observed that for these experiments, the generalization errors of MLP and Transformer are well approximated by the Boolean influence; while MLP-Mixer has higher generalization error. The

---

[10]This is problem dependent; nonetheless, we generally refer to errors of order of magnitude $10^{-2}$ or less.

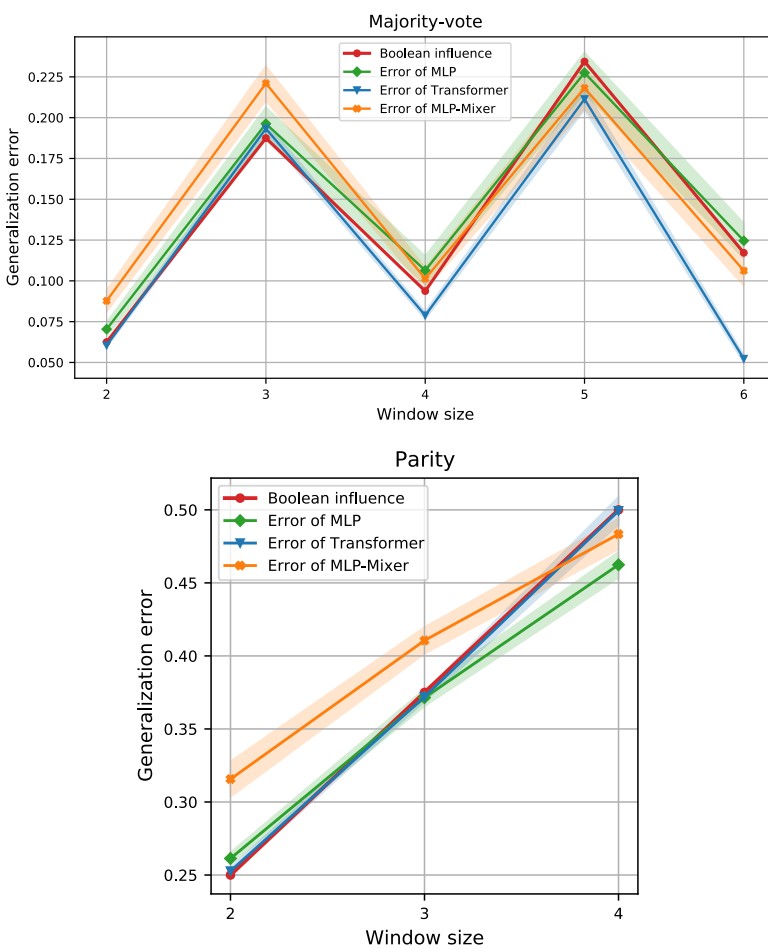

Figure 5: PVR tasks with 3 pointer bits and varying window sizes where the aggregation function is majority-vote (top) and parity (bottom). X-axis represents the window size of the PVR task, and y-axis shows the value of the generalization error and the Boolean influence.

results of Figure 5 and Table 1 indicate that the implicit bias towards low-degree monomials also exists when Adam is used as the optimizer and therefore is not limited to SGD.

Table 1: Generalization error for PVR tasks with 4 pointer bits

| PVR task | | | Generalization error | | |
|---|---|---|---|---|---|
| Aggregation | Window size | Boolean influence | MLP | Transformer | MLP-Mixer |
| Min | 2 | 0.0625 | $0.062 \pm 0.004$ | $0.068 \pm 0.006$ | $0.118 \pm 0.016$ |
| Parity | 3 | 0.1875 | $0.206 \pm 0.004$ | $0.198 \pm 0.015$ | $0.329 \pm 0.017$ |
| Majority | 3 | 0.09375 | $0.099 \pm 0.004$ | $0.095 \pm 0.001$ | $0.194 \pm 0.022$ |
| Majority | 4 | 0.046875 | $0.051 \pm 0.004$ | $0.049 \pm 0.002$ | $0.094 \pm 0.019$ |

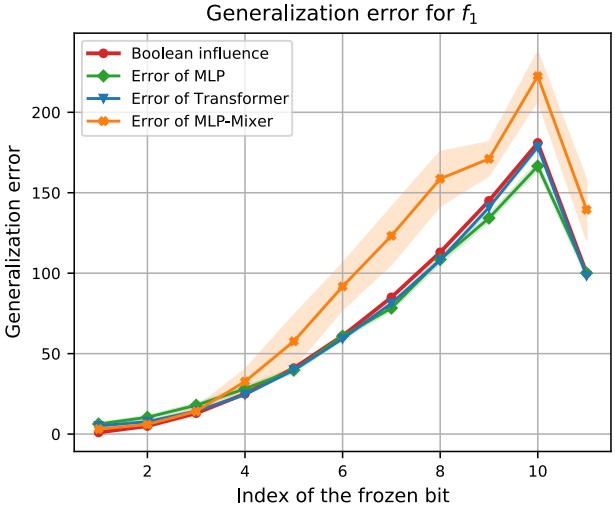

Figure 6: Comparison between the Boolean Influence and generalization error for $f_1(x_1, \ldots, x_{11}) = x_1x_2 + 2x_2x_3 + 3x_3x_4 + \cdots + 10x_{10}x_{11}$. Frozen coordinates are represented by the x-axis; while the y-axis represents the value of generalization error and the Boolean influence.

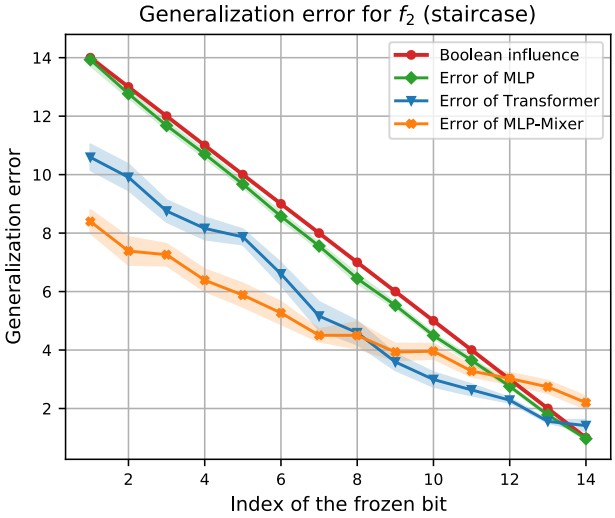

Figure 7: Comparison between the generalization loss in the canonical distribution shift setting and the Boolean Influence for $f_2(x_1, x_2, \ldots, x_{14}) = x_1 + x_1x_2 + x_1x_2x_3 + \cdots + x_1x_2x_3 \cdots x_{14}$.

**Non-PVR examples.** We also experimented on non-PVR functions. As the first example, we consider the target function $f_1(x_1, \ldots, x_{11}) = x_1x_2 + 2x_2x_3 + 3x_3x_4 + 4x_4x_5 + \cdots + 10x_{10}x_{11}$ which is a sum of second degree monomials. For each of the architectures, we freeze a coordinate (ranging from 1 to 11), train the model on the frozen samples using mini-batch SGD and evaluate the generalization loss. The relation between the Boolean influence and the averaged generalization error over 20 runs for $f_1$ is demonstrated in Figure 6. It can be seen that the generalization errors of the MLP and the Transformer model are again well approximated by the Boolean influence. However, the generalization error of the MLP-Mixer follows the trend of Boolean influence with an offset. This implies that the MLP and Transformer have a stronger preference for low-degree monomials than the MLP-Mixer in this case. As the last example, we consider the vanilla staircase

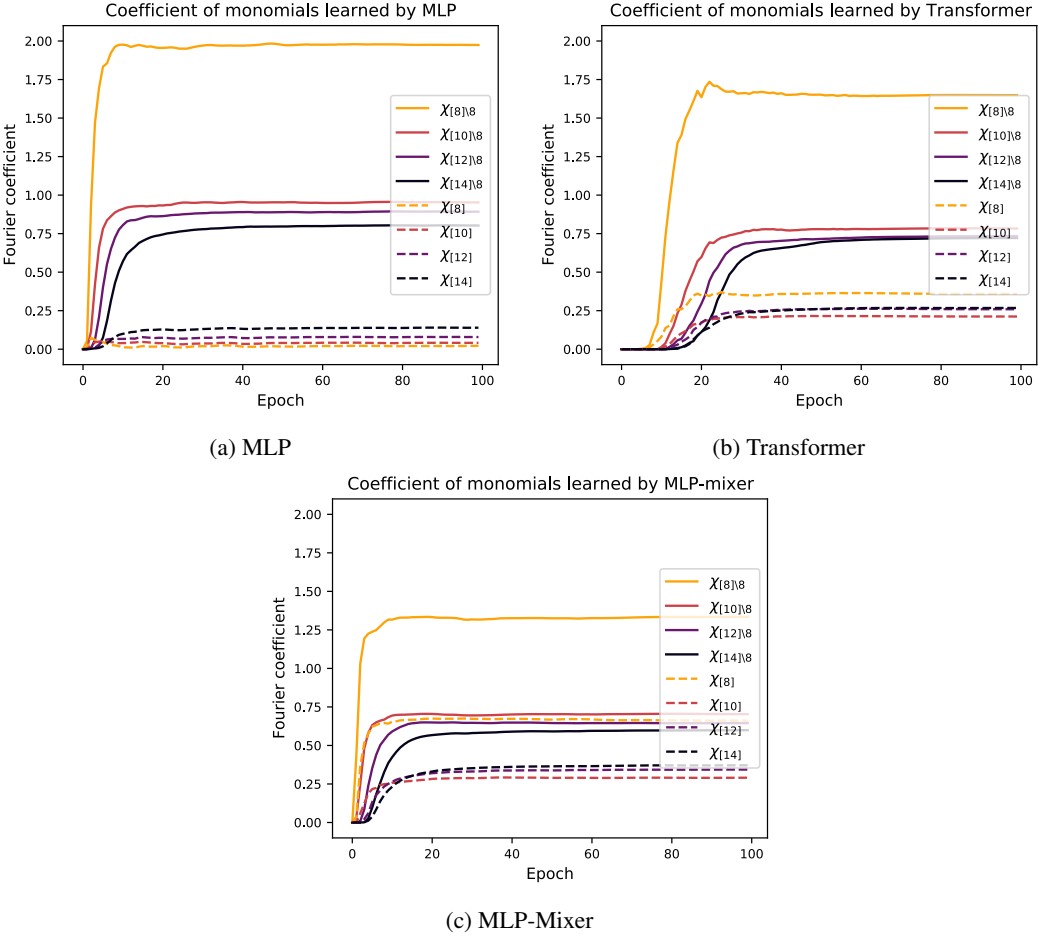

(a) MLP

(b) Transformer

(c) MLP-Mixer

Figure 8: The coefficients of different monomials learned by the MLP, Transformer, and MLP-Mixer when learning the staircase function, with $x_8 = 1$ frozen during the training. Note that $\chi_T(x) = \prod_{i \in T} x_i$, e.g., $\chi_{[8]} = x_1 x_2 \cdots x_8$. Generally, monomials of lower degrees are learned faster by the models. Consequently, the models prefer to learn the monomials which exclude the frozen index.

function for 14 bits, i.e., $f_2(x_1, x_2, \ldots, x_{14}) = x_1 + x_1 x_2 + x_1 x_2 x_3 + \cdots + x_1 x_2 x_3 \cdots x_{14}$ (see [ABAB+21, ABAM22] for theoretical results on such staircase functions). We train models for this function using mini-batch SGD. In Figure 7, we report the generalization errors of the MLP, Transformer, and MLP-mixer models for each frozen coordinate of $f_2$, as well as the values of the Boolean influence of the corresponding index. Note that the generalization errors have been averaged over 20 runs. It can be observed that the generalization loss of MLP is very close to the Boolean influence in this case as well. However, the generalization errors of the Transformer and MLP-Mixer follow the Boolean influence with an offset. It is worth noting that the previous two functions are quite different than the PVR function: the PVR function has strong 'symmetries' given by the fact that each window is treated similarly with the aggregation function, and thus one may expect that certain architectures would exploit such symmetries. Thus, the PVR function is still a staircase function [ABAM22], of leap 2 in the example of Section 1.2, but it is a staircase function with the related symmetries. Instead, the two functions considered here are staircase functions that do not have any such symmetries.

Figure 7 shows that in some cases the Boolean influence may not always give a tight characterization of the generalization error. However, it appears that in such cases the offset still maintains the general trend of the influence. As an attempt to better understand this offset, recall the relation between the

Boolean influence and the generalization error in terms of the implicit low-degree bias: the stronger the preference for low-degree monomials is, the closer the generalization error is to the Boolean influence. We thus plot the coefficient of different monomials for $f_2$ and these three models while $x_8 = 1$ is frozen during the training in Figure 8. One can observe that for this staircase function, the bias towards low-degree monomials is stronger for MLP and it is weaker for Transformer and MLP-mixer. This well explains the relation between the Boolean influence and the generalization error of different models depicted in Figure 7, where the generalization error of MLP is significantly closer to the Boolean influence, compared to the generalization error of Transformer and MLP-Mixer.

## G    Intuition on the linear neural networks

At last, we provide heuristic justifications for the effect of depth and initialization on the generalization error and its closeness to the Boolean influence in the case of linear neural networks. Let $f_{NN}(x; \Theta) = w_L^T(W_{L-1}^T(\cdots(W_1^T x + b_1)\cdots) + b_{L-1}) + b_L$ be a linear neural network with depth $L$, after training in the canonical holdout setting where the $k$-th bit is frozen to 1. Assume the target function to be linear. After training, the neural network learns the frozen function $f_{-k}(x) = f(x_{-k})$. Note that the bias of the frozen function is $\hat{f}(\{\emptyset\}) + \hat{f}(\{k\})$ (where with $\hat{f}$ we denote the Fourier coefficients of the target function $f$), that is expressed by the neural network by the following:

$$B := (w_L^T W_{L-1}^T \cdots W_2^T b_1 + w_L^T W_{L-1}^T \cdots W_3^T b_2 + \cdots + w_L^T b_{L-1} + b_L) + w_L^T W_{L-1}^T \cdots W_2^T w_{1,k}^T \tag{89}$$

where by $w_{1,k}$ we indicate the weights in the first layer of the frozen dimension $k$. Assuming the neural network has learned the function, we have

$$\hat{f}_{NN}(\{i\}) = \hat{f}(\{i\}) \qquad \text{for all } i \neq k, \tag{90}$$

$$\hat{f}_{NN}(\{\emptyset\}) + \hat{f}_{NN}(\{k\}) = B = \hat{f}(\{\emptyset\}) + \hat{f}(\{k\}), \tag{91}$$

where we denoted by $\hat{f}_{NN}$ the Fourier coefficients of $f_{NN}$. Therefore, applying Parseval identity we find that the generalization error equals

$$\text{gen}(f, f_{NN}) = \frac{1}{2}\mathbb{E}_X(f(X) - \hat{f}_{NN}(X))^2 \tag{92}$$

$$= \frac{1}{2}\left(\hat{f}(\{\emptyset\}) - \hat{f}_{NN}(\{\emptyset\})\right)^2 + \frac{1}{2}\left(\hat{f}(\{k\}) - \hat{f}_{NN}(\{k\})\right)^2 \tag{93}$$

$$= \frac{1}{2}\left(\hat{f}(\{\emptyset\}) - (w_L^T \cdots W_2^T b_1 + \cdots + b_L)\right)^2 + \frac{1}{2}\left(\hat{f}(\{k\}) - w_L^T \cdots W_2^T w_{1,k}^T\right)^2 \tag{94}$$

$$= (\hat{f}(\{k\}) - w_L^T \cdots W_2^T w_{1,k}^T)^2. \tag{95}$$

Therefore, the amount of bias captured by $w_L^T W_{L-1}^T \cdots W_2^T w_{1,k}^T$ determines the generalization error. Particularly, if $w_L^T \cdots W_2^T w_{1,k}^T$ goes to zero, the generalization error will become equal to the Boolean influence. Note that $x_k = 1$ during the training, therefore $w_{1,k}$ has the same training dynamics as the bias of the first layer $b_1$.

**Effect of depth.**    From (89), we note that there are $L + 1$ terms that contribute to $B$, and one of them is indeed $w_L^T \cdots W_2^T w_{1,k}^T$. Therefore as the depth $L$ increases, if those terms are appropriately aligned, one can expect that the contribution of each term, including $w_L^T \cdots W_2^T w_{1,k}^T$, decreases; thus, the generalization error becomes closer to the Boolean influence.

**Effect of initialization.**    The gradients of the parameters for a sample $x$ are given by

$$\nabla_{b_L} L(\Theta, x, f) = (f_{NN}(x; \Theta)(x) - f_{-k}(x)),$$
$$\nabla_{b_{L-1}} L(\Theta, x, f) = (f_{NN}(x; \Theta)(x) - f_{-k}(x))w_L,$$
$$\vdots$$
$$\nabla_{b_1} L(\Theta, x, f) = (f_{NN}(x; \Theta)(x) - f_{-k}(x))W_2 W_3 \cdots w_L.$$

Now, consider the first update of the parameters. As we decrease the scale of initialization, the ratio of $\frac{\nabla_{b_L}}{\nabla_{b_{L-1}}}, \cdots, \frac{\nabla_{b_2}}{\nabla_{b_1}}$ increases which implies that $b_1$ would have the smallest update and $b_L$ will have the largest update. Since the dynamics of $w_{1,k}$ and $b_1$ are the same, the frozen dimension would contribute the least to the bias after the first iteration. Our experiments on decreasing the scale of initialization suggest that this argument is not limited to the first iteration. In other words, using small enough initialization the bias will be mostly captured by the bias terms in other layers, which results in generalization error being close to the Boolean influence.