# OpenReview forum: "Learning to Reason with Neural Networks: Generalization, Unseen Data and Boolean Measures"
_NeurIPS.cc/2022/Conference — NeurIPS 2022 Accept_

### Official Review · Reviewer_PXDA · 2022-07-11

**Rating:** 7
**Confidence:** 3
**Soundness:** 3 good
**Presentation:** 3 good
**Contribution:** 3 good

**Summary:**

The paper uses the Pointer Value Retrieval (PVR) benchmark to study the problem of reasoning with DNNs. In the PVR problem, MNIST digits in the string format are provided as input (say restricted to 0,1 digits) and the task is to perform some logical operation over these to find the final label. The paper focuses on the string version of the problem (rather than matrices).  The benchmark helps study the tradeoff between memorization and reasoning by acting with a distribution shift at testing. This paper focuses on the logical/Boolean learning component of the challenge.

**Questions:**

 The presentation starts with a more general problem of reasoning over DNN before narrowing to PVR. It then further narrows to PVR over string. It then narrows to the Boolean learning component. It finally narrows to canonical holdout. This is a very odd presentation choice and makes the reader assume the breadth of the paper's contribution to be initially much larger.  Is there anyway, the initial presentation (first 2.5 pages) can be rewritten to better reflect the key contributions of the paper?

**Limitations:**

There are no potential negative societal impact of this work to the best of this reviewer's understanding.

**Strengths And Weaknesses:**


+ The paper addresses an important challenge of reasoning with DNNs and studies the problem using a variant of PVR benchmark.

+ The paper shows that if the target function is highly noise unstable - stability decreases faster than any inverse polynomial in n, then gradient descent will not learn the 2n extension of the function in polynomial time and with a polynomially sized NN. So, noise stability gives a proxy to generalization error.

+ The theoretical result is complemented with empirical evaluation over architectures such as MLPs and transformers.

- Boolean function learning has been extensively studied in formal methods and AI. The O'D14 reference from the paper is an excellent background, and there have been some practical algorithms for special classes of Boolean function that might be of interest to the authors such as https://ieeexplore.ieee.org/abstract/document/8849233, https://link.springer.com/article/10.1007/s10817-018-9499-8, https://eprint.iacr.org/2021/1099 Discussion of this larger literature on Boolean function learning would be useful.

---

> ### Author Response · Authors · 2022-08-02
> **Response to Reviewer PXDA**
>
> We thank the reviewer for the constructive comments. We address the remarks and questions in the list below.
>
> > * Q1: Practical algorithms for special classes of Boolean function that might be of interest to the authors such as \url{https://ieeexplore.ieee.org/abstract/document/8849233},\url{https://link.springer.com/article/10.1007/s10817-018-9499-8}, \url{https://eprint.iacr.org/2021/1099} Discussion of this larger literature on Boolean function learning would be useful.
>
> A1: We thank the reviewer for pointing out the references. We included these references in our paper. We will look into further related works, and include them in a larger literature discussion in the camera-ready.
>
> > * Q2: The presentation starts with a more general problem of reasoning over DNN before narrowing to PVR. It then further narrows to PVR over string. It then narrows to the Boolean learning component. It finally narrows to canonical holdout. This is a very odd presentation choice and makes the reader assume the breadth of the paper's contribution to be initially much larger. Is there anyway, the initial presentation (first 2.5 pages) can be rewritten to better reflect the key contributions of the paper?
>
> A2: We believe that there are arguments in favor of the current abstraction flow and arguments against. What is in favor is that our theoretical results apply to general Boolean functions and not just PVR, making the more general Boolean setting relevant. Regarding the implicit bias towards low-degree monomials, we also did experiments for other Boolean functions than PVR and the general trend observed for PVR seems to be present (Figures 6 and 7). However, we emphasize that the phenomenon that the Boolean influence is a tight approximation for generalization is particularly observed for PVR functions (Figure 2) and indicates that the models considered (MLPs and Transformers) fail to reason in these PVR tasks with canonical holdout.
> Nonetheless, the reviewer is correct that the canonical holdout is a specific type of holdhout/unseen data model, and here we can be more upfront in the introduction about it. We revised the introduction to make terms like "unseen data" and "distribution shift" more clear. We will implement further clarifications in the camera-ready (where page limit is extended).

---

> > ### Comment · Reviewer_PXDA · 2022-08-10
> > **Thank you.**
> >
> > Given the response of the authors, the reviewer is raising the score.

---

> > > ### Author Response · Authors · 2022-08-10
> > > **Thanks**
> > >
> > > We thank the reviewer for appreciating our comments and raising the score.

---

### Official Review · Reviewer_mpbM · 2022-07-11

**Rating:** 7
**Confidence:** 4
**Soundness:** 3 good
**Presentation:** 3 good
**Contribution:** 3 good

**Summary:**

The paper develops a combination of empirical and theoretical analyses for the recently introduced pointer value retrieval (PVR) benchmark, proposed to measure how a neural network reasons about a logical task. The paper has two main results which connects generalization error of a learner performing gradient descent on l2-norm loss in a (pseudo) Boolean logical task to two measures in Boolean optimization under settings where a shift between the distribution of training and test sets exists or can be avoided:

In order to learn general logical functions (Boolean and pseudo-Boolean functions) using gradient descent, it is theoretically proven that the learner incurs a generalization error which is lower-bounded by the noise stability of the target function (i.e., how much the output of the function changes on average as the result of Boolean perturbations). The analysis is distinct from the prior theoretical work in that it uses a measure that poses a property over the target function rather than the function class to be learned (or an alignment between the two).

In the distribution shift setting (and focusing on a particular shift called canonical holdout which fixes a binary coordinate in the training set), the generalization error of the learner with gradient descent follows an intuitive measure called Boolean influence. The study further establishes the existence of a bias in neural networks towards learning low-degree polynomials (low frequencies). This result is theoretically proven for the linear models and empirically justified for some neural networks such as MLP and Transformers.


**Questions:**

Implication of the theoretical results to the problem of reasoning in neural networks under PVR. The paper is motivated by the PVR benchmark and puts forward a theoretical analysis for that under distribution shift. Discussing the implications of these theoretical results to the reasoning problem (under PVR) can further strengthen the paper. In a high level (if I am correct) the paper seems to  suggest that for the Boolean problems the neural network choses a solution (logical function) which has a bias towards low-degree polynomials (natural given that the known empirical biases of neural networks towards low-frequencies components in other domains). While this bias seems to be beneficial in several cases in terms of generalization (e.g., when there is no distribution shift), the bias does not seem to be the desired one in general (e.g., when there is a distribution shift) especially in the logical tasks involved in the Boolean-PVR benchmark. A symmetry-compensation bias, however, which is either explicitly-imposed via regularization or implicitly-imposed via a new architecture, seems to be what is required (as alluded in the future work). Does this imply that learning gradient descent on the l2 norm loss over MLP and Transformers lack such symmetries and are thus not well-equipped for reasoning tasks such as the one in PVR?

Noise stability analysis of the PVR task. To complete the exposition, is it straightforward to find the \delta-noise stability term for the logical PVR function considered as a function of the problem dimension n and the window size w analytically? Is it a monotone function of w?

Analysis of distribution shift for the nonlinear case. The analysis of the distribution shift with canonical holdout is done for the linear case. What are the challenges to do the analysis for the nonlinear case? Can the authors elaborate more on Remark 3—is the suggested analysis trivial?

MLP-Mix. More insights on why there is no bias towards low-degree solutions in the MLP-Mix case would be beneficial.  Also more details on how the architecture is adapted to the Boolean problem (given that MLP-Mix is developed for images) helps the reader better contrast the three architectures (MLP, MLP-Mix, and Transformers) employed in the work and understand the differences in their behaviors on the PVR benchmark.

Background on low-degree bias. While the analysis for the bias of neural networks towards low-degree polynomials is novel from the theoretical perspective the empirical evidences are observed in prior works in applied machine learning domains involving Boolean objects (e.g., analysis of fitness functions in biology) which might be worth mentioning, e.g. see "Epistatic Net allows the sparse spectral regularization of deep neural networks for inferring fitness functions." Nature communications 12.1 (2021): 1-10.

Uniform assumption. Throughout the paper the assumption on the non-hold-out indices is a Bernouli distribution. In practice this might be too restrictive; it is common that the training data is concentrated around a point in the hamming space in many learning scenarios. While it is fairly natural for the theoretical analysis to assume uniformity, from a practical point of view do the authors expect the results to hold under a non-uniform distribution case?

**Limitations:**

Discussed before.

**Strengths And Weaknesses:**

Overall the paper investigates a new and timely problem of reasoning in neural networks from a theoretical perspective with clear results and contributions which help advance both machine learning research and Boolean optimization—an intersection with growing research problems. Connecting generalization performance of logical functions to Boolean measures is novel, intuitive, and interesting. It also opens up new research questions to be addressed in learning Boolean functions. I have a few questions and suggestions for the authors mostly to improve the paper.

---

> ### Author Response · Authors · 2022-08-02
> **Response to Reviewer mpbM**
>
> We thank the reviewer for the constructive comments. We address the remarks and questions in the list below.
>
> > * Q1: Does the low-degree bias imply that learning using GD on the l2 loss over MLP and Transformers lack of symmetry-compensation and are thus not well-equipped for reasoning tasks such as the one in PVR (under distribution shift)?
>
> A1: Note that all models have some symmetries in the beginning. However, at the end, low-degree implicit bias wins over the symmetry. In other words, architectures such as MLP and the Transformer, are implicitly biased towards low-degree solutions rather than symmetric solutions. Thus, we can say that they are not well-equipped for reasoning on PVR tasks under the canonical holdout setting. For learning to reason, we need to have models that learn the ‘simplest’ / most ‘symmetric’ representation (these terms are not necessarily well-defined across different tasks). In the future, we would like to find a method (such as via regularization) that encourages models to learn these kinds of representations rather than the lowest-degree representation.
>
> > * Q2: Noise stability analysis of the PVR task. To complete the exposition, is it straightforward to find the $\delta$-noise stability term for the logical PVR function considered as a function of the problem dimension n and the window size w analytically? Is it a monotone function of w?
>
> A2: Note that for the non-overlapping windows case considered in Appendix D1, we have that $n = p+2^pw \asymp 2^pw$. Thus, replacing $p = \log(n/w) $ in (80) one can get $Stab_\delta(f) \asymp (1-\delta)^{\log n+w -\log w} + (1-\delta)^{\log(n/w)}(1-(1-\delta)^w) \cdot  Stab_\delta(g) $. The monotonicity with respect to $w$ depends on the choice of the aggregation function $g$, however for many common aggregations it is decreasing with $w$. For instance, if the aggregation function is the parity function ($g(x_1,...,x_w) = \prod_{i=1}^w x_i$), one can observe that $Stab_\delta(g) = (1-2\delta)^w$. Then, eq (80) becomes $Stab_\delta(f) = (1-\delta)^{w+p} [ 1-(1-2\delta)^w] +(1-\delta)^p $, and $Stab_\delta(f)$ is decreasing with $w$. Also, if $g$ is the majority vote function $g(x_1,...,x_w) = sign(\sum_{i=1}^w x_i)$, then $Stab_\delta (g) = 1-2/\pi \cdot  \arccos(1-2\delta)$ (see e.g., [O’D14]). Plugging this in (80), one can observe that also for majority vote $Stab_\delta(f) $ is decreasing with $w$. We will complete the exposition including details and examples in the camera-ready version.
>
> > * Q3: The analysis of the distribution shift with canonical holdout is done for the linear case. What are the challenges to do the analysis for the nonlinear case? Can the authors elaborate more on Remark 3- is the suggested analysis trivial?
>
> A3:  Remark 3 is, in our opinion, a promising strategy for analyzing this phenomenon in the non-linear setting. In fact, in [ABAM’22] the authors prove that SGD on a two layers fully connected neural network in the mean field regime learns features hierarchically: e.g. if the data is generated by a Boolean staircase $f(x) = x_1+x_1x_2+x_1x_2x_3+\cdots +x_1x_2\cdots x_k$ (where $x_i \sim \mathrm{Unif}({\pm1\})$) the network learns first the lowest degree term $x_1$, and then learns the other terms sequentially. We believe that one could prove a similar dynamics in the holdout setting (this would require adding a bias term to their architecture), and consequently prove the Boolean influence tightness. Nevertheless, we expect this extension to be non-trivial (probably about 10 pages of computations) and it is left as a future direction.

---

> > ### Author Response · Authors · 2022-08-02
> > **Response to Reviewer mpbM (continuation)**
> >
> > > * Q4: MLP-Mix. More insights on why there is no bias towards low-degree solutions in the MLP-Mix case would be beneficial. Also more details on how the architecture is adapted to the Boolean problem (given that MLP-Mix is developed for images) helps the reader better contrast the three architectures (MLP, MLP-Mix, and Transformers) employed in the work and understand the differences in their behaviors on the PVR benchmark.
> >
> > A4: This is a good question. Notice first that we cannot say that there is “no” bias towards low-degree, as there is still preference towards the low-degree monomials than the higher ones (see Fig 8c). What is true is that that preference is milder in the case of MLP-Mixers than MLP or Transformers. Further, the generalization error seems to still follow the trend of the Boolean influence, but with an offset. At this point we only have speculations about this offset and do not have anything conclusive to share; we hope to have more in the future. Notice however that our priority has been to figure out how to counter this low-degree bias, with regularizations or architecture constraints. Here the difficulty is to promote symmetries in the solution, without handcrafting these explicitly by exploiting prior knowledge on the problem. It appears that defining a proper cost function for the function description would be the most effective, if the latter can be implemented efficiently.  Regarding the MLP-Mixers adaptation question, note that although they were originally designed for images, they actually partition images into "patches" and treat the image as a sequence of "patches". Therefore, apart from the first layer that maps an RGB patch to a fixed-sized vector (the patch embedding), the rest of the architecture is agnostic to the nature of the inputs. For example, to use it in PVR or any sequence learning problems with fixed-length sequence, we simply replace the first layer (patch embedding) with a token embedding.
> >
> > > * Q5: Background on low-degree bias. While the analysis for the bias of neural networks towards low-degree polynomials is novel from the theoretical perspective the empirical evidences are observed in prior works in applied machine learning domains involving Boolean objects (e.g., analysis of fitness functions in biology) which might be worth mentioning, e.g. see "Epistatic Net allows the sparse spectral regularization of deep neural networks for inferring fitness functions." Nature communications 12.1 (2021): 1-10.
> >
> > A5: We thank the reviewer for pointing out this reference. We included it in our paper. We will look into further related works, and include them in a larger literature discussion in the camera-ready.
> >
> > > * Q6: Uniform assumption. Throughout the paper the assumption on the non-hold-out indices is a Bernoulli distribution. In practice this might be too restrictive; it is common that the training data is concentrated around a point in the hamming space in many learning scenarios. While it is fairly natural for the theoretical analysis to assume uniformity, from a practical point of view do the authors expect the results to hold under a non-uniform distribution case?
> >
> > A6: If the sampling of the bits is independent (with the same distribution or not), we can handle it. In fact, the Fourier-Walsh transform can be defined in a non-uniform input basis, and one could define the Noise Stability and Influence from the coefficients in the new basis. However, if the bits are dependent, this requires new investigations. We note that for specific input distributions, e.g., jointly Gaussian data, one could use Hermite basis functions and whitening steps, but generally this is a non-trivial extension.

---

> > > ### Comment · Reviewer_mpbM · 2022-08-07
> > > **Thanks**
> > >
> > > Thanks for the response. I have read it and do not have further questions from the authors.

---

### Official Review · Reviewer_g1sx · 2022-07-12

**Rating:** 6
**Confidence:** 2
**Soundness:** 3 good
**Presentation:** 2 fair
**Contribution:** 3 good

**Summary:**

The paper presents some iinteresting results on the properties of NNs on boolean function learning, that are supported by experimental evaluation,

**Questions:**

Why use a quadractic error measure?

You use 3 architectures. Please explain why these three.

Fig 3 results seem to be designed to explain the mixer results? Maybe explain that from the start.

how would the classifiers work if there is no noise?

What is  " the empirical claim made in [ZRKB21]." ?

**Limitations:**

The authors discuss some extensions to their setting, but that is mostly done n the Appendixes, and shhow some clear research diretcions in the Conclusions.

**Strengths And Weaknesses:**

I liked the paper, but I had  to overcome two initial problems:
Title is really misleading, why is this about learning to reason?
Structure: I found sec 1 jumps everywhere, namely   1,,2 present results (boolean influence) wIthout a def, (see 3.1 for more details?).
def 2 is only used in sec 2?
Why: 1.3 Further related literature, further?

Finally the exp design must be in the main text. Especially when your results are impressive, the strength of this paper is in the match between theory and results.

---

> ### Author Response · Authors · 2022-08-02
> **Response to Reviewer g1sx**
>
> We thank the reviewer for the constructive comments. We address the remarks and questions in the list below.
>
> > * Q1: Title is really misleading, why is this about learning to reason?
>
> A1: The title `learning to reason’ is motivated by the focus on the PVR benchmark, which is relevant for evaluating the ability to reason in learning algorithms (as pointed out in [ZRKB’21]). More specifically, in PVR tasks we say that the algorithm learns to reason (instead of just memorizing the data) if it outputs the correct function despite some data holdout during training. In fact, the algorithm has the possibility to learn the pointer rule and the aggregation function, from the windows where data is not withheld. Although our theoretical results hold for generic Boolean functions, and are not specific to PVR tasks, we believe that the surprising tightness of the influence characterization in PVR tasks in a specific holdout setting (observed in Figure 2) gives insights into the ability to reason in Transformers and MLPs, which is what motivates our title. More precisely, the generalization loss (under canonical holdout) for such models is well-approximated by the Boolean influence. Hence, it shows that these architectures fail to reason (i.e. they do not have small enough generalization error) on PVR tasks under the canonical holdout setting.
>
>  > * Q2: Structure: I found sec 1 jumps everywhere, namely 1.2 present results (Boolean influence) wIthout a def, (see 3.1 for more details?). def 2 is only used in sec 2? Why: 1.3 Further related literature, further?
>
> A2: Thank you. We addressed these points in the revised version. Specifically, we included the definition of Fourier-Walsh expansion and Boolean influence in Section 1.2 and we removed ``further’’ from the title of section 1.3. We did not move the definition of canonical holdout, as it is used in section 1.2.
>
> > * Q3: The exp design must be in the main text. Especially when your results are impressive, the strength of this paper is in the match between theory and results.
>
> A3: Thank you. We included some details of the implementation for the Transformer and MLP-Mixer in the main body. We will include a more detailed description in the camera ready version (where page limit is extended).
>
> > * Q4: Why use a quadratic error measure?
>
> A4: The quadratic loss is particularly convenient for theoretical analysis and simplifies the maths. In fact, we are able to use Parseval’s theorem and make the connection with the Fourier domain. The 0/1 loss (accuracy) for tasks with Boolean output is directly connected to the L2 loss and it can be directly explained by the Boolean influence. However, for other loss functions (such as cross-entropy loss used for training on classification tasks) the connection is more tedious (i.e., the resulting expression will not be as simple to state, although the insights are similar we believe).
>
> > * Q5: You use 3 architectures. Please explain why these three.
>
> A5: We focused on the architectures that were used in the original PVR paper [ZRKB’21] to compare the results, since our primary aim was to provide a theoretical analysis for their empirical observations. Nonetheless, we believe those provide a good coverage of common neural network architecture backbones used nowadays (except for convolutional networks, which are typically used only with image / video inputs): MLP is the most vanilla fully connected neural network architectures; Transformers use many modern components such as attention, layer normalization, and residual connections, and are widely used in all major NLP applications; MLP-Mixers were originally designed for vision problems but they can be easily applied to any (fixed-length) sequence learning problems, and represent a new family of emerging architectures that does not use either attention or convolution.

---

> > ### Author Response · Authors · 2022-08-02
> > **Response to Reviewer g1sx (continuation)**
> >
> > > * Q6: Fig 3 results seem to be designed to explain the mixer results? Maybe explain that from the start.
> >
> > A6: Figure 3 aims to demonstrate the relation between the bias towards low-degree monomials and having the Boolean influence as the generalization error. We observe that neural networks can learn each monomial of the target function by either including or excluding the frozen bit. Particularly, if they have bias towards low-degree monomials, they will learn the monomials which exclude the frozen bit. Consequently, we would have the Boolean influence as the generalization loss. As depicted in Figure 3 (left), MLPs have a strong preference towards low-degree monomials, and prefer to learn each of the monomials by excluding the frozen variable (solid lines) rather than including it (dashed lines). Hence, the generalization error for the MLP model is well-approximated by the Boolean influence. In contrast, the MLP-Mixer (Figure 3, right) has a weaker preference for low-degree monomials. For instance, the value for $x_6$ is non-negligible (the model could have completely used the lower-degree monomial $1$ instead). This non-negligible value for monomials including the frozen coordinate results in a larger gap between the generalization error and the value of the Boolean influence. We will add a clarification about Figure 3 in the camera ready version (where page limit is extended).
> > Finally, note that the same plot for the Transformer model is depicted in Figure 9. Also, Figure 8 does a similar comparison for a vanilla staircase function.
> >
> >
> > > * Q7: How would the classifiers work if there is no noise?
> >
> > A7: In the proof of Theorem 1 we need the noise as a smoothing parameter. We remark that such noise can be small (e.g., it can be vanishing with n). For this reason, we do not expect the result to break down without the noise, it is rather a technicality that makes the proof simpler.
> >
> > > * Q8: What is ``the empirical claim made in [ZRKB21]." ?
> >
> > A8: In [ZRKB’21] the authors empirically observe that the difficulty of learning a target function with gradient descent on neural networks increases with the noise sensitivity of the target ([ZRKB’21] Section 4.1 and Appendix C). We clarified this in the revised version of the paper.

---

### Official Review · Reviewer_Mz9k · 2022-07-12

**Rating:** 5
**Confidence:** 3
**Soundness:** 2 fair
**Presentation:** 3 good
**Contribution:** 2 fair

**Summary:**

This paper mainly focuses on learning logical function with GD and investigates generalization error under matched and mismatched setting. For GD generalization error under matched setting, the authors prove a lower-bound related especially to noise-stability, supporting a formal conjecture. Under the canonical holdout, the generalization error is close to Boolean influence for relevant architecture, which puts forward an interesting ‘low-degree implicit bias hypothesis’.

**Questions:**

Experiments conducted are under the distribution shift setting. Is it necessary to carry out experiments verifying the generalization error lower bound under the matched setting and observe difference by changing noise stability?

**Limitations:**

Three architectures, MLPs, Transformers and linear models, are considered for experiments. It is worthy to consider an extension to other architectures and prove the Boolean influence tightness, as mentioned in the future pursuits.
Unseen Data, appeared in the title, is not mentioned except in the future pursuits. Maybe there should be more explanations about it.

**Strengths And Weaknesses:**

Overall, the compact structure explains points clearly and logically. Introduction part basically and vividly explains the main contributions of this article, providing several instances for easy-understanding(e.g. illustration of low-degree bias hypothesis). After that, matched and mismatched setting are separately illustrated. Then, several experiments on MLPs, Transformers and linear models provided effectively support the formal points and hypothesis.
The idea that introducing PVR benchmark to understand the limits of deep learning on tasks  and investigating ‘memorization and reasoning’ trade-off is practical and can be used in many domains. The ‘canonical’ holdout that a single feature/bit is frozen at training is also an important case. This paper exactly considers the two settings, proving a lower-bound and a tightness, which other papers haven't investigated.

---

> ### Author Response · Authors · 2022-08-02
> **Response to Reviewer Mz9k**
>
> We thank the reviewer for the constructive comments. We address the remarks and questions in the list below.
>
> > * Q1: Experiments in the matched setting (generalization error and noise stability).
>
> A1: We did not redo experiments as they were made available in the first PVR paper ([ZRKB’21] Section 4.1 and Appendix C) and as our contribution here was to obtain a theoretical result providing a lower-bound that degrades with the noise-sensitivity of a function, in agreement with the experiments done in [ZRKB’21]. Note however that the noise sensitivity may not give a tight characterization of the generalization error. In fact, we believe that the noise-sensitivity captures a necessary condition, i.e., too much sensitivity implies failure of learning, but it is unlikely that a comparable sufficient condition holds.
>
>
> > * Q2: Boolean influence tightness for other architectures.
>
> A2: Further investigation on other architectures would naturally add to the picture; some is considered for future work but there is generally no clear cutoff on where to stop. In this paper, we push forward an encouraging first step that the Boolean influence is a good approximation for important classes of architectures. We studied models such as MLPs, MLP-mixers and Transformers as they were the main architectures used in the original PVR paper [ZRKB’21] and as they represent important classes. We refer to the answer to Q5 of Reviewer g1sx for further comments on the choice of architectures. On the theoretical level, we expect the proof for the general case to be significantly harder; currently the theory that allows to study the dynamics of GD tightly is limited to specific (regular) architectures even in the standard matched setting, let alone questions related to holdout or distribution shift.
>
>
> > * Q3: Unseen Data, appeared in the title, is not mentioned except in the future pursuits. Maybe there should be more explanations about it.
>
> A3: By "unseen data" we mean the canonical holdout setting, as in Definition 2. The main motivation to choose "unseen data" in the title was to keep it simple. We did not want to use "canonical holdout" in the title as the term is not defined yet, and thus used a more generic word (note that holdout indeed makes some of the data unseen, so the terminology is appropriate). We added a clarification about this in the revised version.

---

### Meta-Review · Area_Chair_24S5 · 2022-08-26

**Recommendation:** Accept
**Confidence:** Certain

**Metareview:**

The paper addresses in a formal and solid way a specific learning task with the intent to assess the limitations of different neural networks architectures. The addressed task requires some degree of reasoning for the architecture to learn it in a satisfactory way. Although the authors eventually restrict the scope of their work, personally I think the paper gives insights about some aspects of learning bias in specific architectures that are not so evident at first sight, in fact contributing to the creation of a different perspective for the pursuit of a better understanding of deep learning models. In their rebuttal authors have clarified some issues raised by the reviewers, and no specific negative concerns have been raised on the presented work, except for the relatively limited scope of the theoretical analysis and range of applicability to other architectures not covered in the paper. Overall I think the paper contains interesting material and concepts that can be of interest for NeurIPS audience, and harbingers of future developments.

**Award:**

No

---

### Decision · Program_Chairs · 2022-09-14

Accept